



# On the suitability of general-purpose finite-volume-based solvers for the simulation of atmospheric-boundary-layer flow

Beatrice Giacomini[1] and Marco G. Giometto[1]

[1]Department of Civil Engineering and Engineering Mechanics, Columbia University in the City of New York, 500 W 120th St, New York, NY 10027, USA.

**Correspondence:** Marco Giometto (mg3929@columbia.edu)

**Abstract.** In the present work, the quality and reliability of a general-purpose second-order-accurate Finite-Volume-based (FV) solver are assessed in wall-modeled large-eddy simulations of a neutrally-stratified Atmospheric-Boundary-Layer (ABL) flow with no Coriolis effects. The sensitivity of the solution to parameters such as grid resolution and aspect ratio is analyzed, and results are contrasted against those from a well-proven mixed Pseudo-Spectral and Finite-Difference (PSFD) code. Considered

flow statistics include mean streamwise velocity, resolved Reynolds stress, turbulence intensities, skewness, kurtosis, spectra and spatial autocorrelations. It is found that first- and second-order velocity statistics are sensitive to the grid resolution and to the details of the near-wall numerical treatment, and a general improvement is observed with horizontal grid refinement. Higher-order statistics, spectra and autocorrelations of the streamwise velocity, on the contrary, are consistently mispredicted, regardless of the grid resolution. Skewness and kurtosis of the streamwise velocity, for instance, are overpredicted in the surface

layer, whereas one-dimensional spectra feature a strong sensitivity to the grid resolution in the production range and a rapid decay of energy density at higher wavenumber. In addition, the typical signatures of Large-Scale Motions (LSMs) are absent in the premultiplied streamwise velocity spectra, the spatial autocorrelation functions rapidly decay along both the streamwise and spanwise coordinate directions, and instantaneous snapshots of the velocity field are populated by relatively short and thin streaks, confirming that the flow lacks LSMs. Further, the dominant mechanism supporting the tangential Reynolds stress in

ABL flow – spanwise-paired sweeps and ejections– is much weaker than what commonly observed in ABL flows, ejections are severely underpredicted, and sweeps account for most of the tangential Reynolds stress in the surface layer, which is at odds with available measurements and with corresponding results from the PSFD-based solver. The inability of the solver to correctly capture the spatially-localized and relatively strong ejection events, in the authors' opinion, is the root-canse of many of the observed mismatches and sensitivity of flow statistics to grid resolution. The present findings show that truncation errors

have an overwhelming impact on the predictive capabilities of second-order-accurate FV-based solvers, introducing a degree of uncertainty in model results that may be difficult to quantify across applications involving boundary-layer flows. Although mean flow and second-order statistics become acceptable provided sufficient grid resolution, the use of said solvers might prove problematic for studies requiring accurate higher-order statistics, velocity spectra and turbulence topology.



## 1 Introduction

An accurate prediction of Atmospheric-Boundary-Layer (ABL) flows is of paramount importance across a wide range of fields and applications, including weather forecasting, complex-terrain meteorology, agriculture, air quality modeling and wind energy (Fernando, 2010; Whiteman, 2000; Oke et al., 2017; Calaf et al., 2010a; Shaw et al., 2019). There is indeed a rapidly growing interest in these applications, motivated by the increasing need for high-resolution information on turbulence and turbulent transport across scales, for the prediction of severe-weather events (hurricanes, heat waves) and for the design of mitigation strategies against climate change.

Since the early work of Deardorff (1970), the Large-Eddy-Simulation (LES) technique has spurred considerable insight on the dynamics of ABL flows. In LESs, only the motions at large scales are directly resolved on the given grid, often dictated by the available computational resources, whereas contributions from Sub-Grid-Scale (SGS) motions to momentum and mass transport and energy dissipation are parameterized as functions of resolved-scale quantities. In the last few decades, the increased availability and power of high-performance-computing facilities (national supercomputers, cloud-based services, etc.) have led to a proliferation of LES studies of ABL processes. These studies include fundamental analysis of the ABL flow over rough surfaces (Bou-Zeid et al., 2009; Anderson and Meneveau, 2010; Salesky et al., 2017; Momen et al., 2018), over and within plant canopies (Yue et al., 2007b; Chester et al., 2007b; Pan et al., 2014; Chamecki, 2013; Bailey and Stoll, 2013) and urban canopies (Tseng et al., 2006; Bou-Zeid et al., 2009; Cheng and Porté-Agel, 2013; Li et al., 2016b; Giometto et al., 2017; Nazarian et al., 2018; Li and Bou-Zeid, 2019), and the investigation of the ABL flow for wind energy applications (Calaf et al., 2010b; Sharma et al., 2016; Abkar and Porté-Agel, 2013; Stevens and Meneveau, 2017), amongst others.

When it comes to the simulation of ABL flows, fully- or partially-dealiased mixed Pseudo-Spectral- and Finite-Difference- (PSFD-) based solvers have been the go-to approach since the studies of Moin et al. (1978) and Moeng (1984). Such solvers combine the accuracy and efficiency of the Fourier partial-sum representation in the horizontal coordinate directions with the more flexible Finite-Difference approach in the vertical (non-periodic) one. Nowadays, most LESs of ABL flows still rely on PSFD-based solvers (see e.g., Sullivan et al., 1994; Albertson and Parlange, 1999). These solvers are known to yield accurate flow fields up to the LES cut-off frequency and to produce good results when used in conjunction with dynamic SGS models (Germano et al., 1991; Lilly, 1992), even when relying on a low-order Finite-Difference discretization in the vertical coordinate direction. However, single-domain PSFD-based solvers are limited to regular domains and, in general, are not suitable for accurately representing sharp variations in the flow field, such as shocks or gas-solid interfaces. In addition, problematics may arise when parallel computing is attempted (Canuto et al., 2006; Margairaz et al., 2018). With the increasing need to account for complex geometries and multi-physics, several efforts have been devoted to the mitigation of the aforementioned limitations. For example, Fang et al. (2011) and Li et al. (2016a) devised strategies to alleviate the Gibbs oscillations that arise when using Pseudo-Spectral expansions in multiply-connected domains, whereas in Chester et al. (2007a) a fringe-forcing technique was proposed to simulate non-periodic flows within a Fourier-based Pseudo-Spectral solver. A shortcoming of these formulations is that they are often ad-hoc or validated only for specific applications, thus introducing a degree of uncertainty in model results and conservation properties of the numerical scheme that might be hard to quantify and generalize.





There is hence a growing interest in using computational-fluid-dynamics solvers for LES based on compact spatial schemes
(Orlandi, 2000; Ferziger and Peric, 2002). The Parallelized Large-Eddy Simulation model (Raasch and Schröter, 2001; Maronga
et al., 2015) and the Weather Research and Forecasting model (Skamarock et al., 2008; Chen et al., 2011) are prominent ex-
amples of said efforts. Both the approaches are based on a high-order Finite-Difference discretization, whereby a system of
dynamical solvers is combined to simulate a range of meteorological phenomena. The resulting solvers are relatively versatile,
suitable for complex geometries via structured and unstructured meshes, able to support local grid refinement and relatively
straightforward to parallelize, given the compact nature of the spatial discretization. Nonlinear terms are typically approx-
imated by using high-order upwind-biased differencing schemes, which are suitable for LES in complex geometries with
arbitrary grid stretching factors and outflow boundary conditions (Beaudan and Moin, 1994; Mittal and Moin, 1997). Such
schemes, however, are known to be overly dissipative and do not conserve energy. In addition, while satisfactory first- and
second-order flow statistics can be obtained in complex geometries at moderate Reynolds numbers (Mittal and Moin, 1997),
the excessive damping of resolved-scale energy at high wavenumber is likely to compromise their predictive capabilities for
high-Reynolds ABL-flow applications. On the other hand, if central schemes are used instead for the evaluation of nonlinear
terms, no numerical dissipation is introduced, but truncation errors can have an overwhelming impact on the computed flow
field (Ghosal, 1996; Kravchenko and Moin, 1997), especially in simulations where the grid is just fine enough to resolve the
large-scale flow structures. These limitations typically result in a strong sensitivity of the solution to properties of the spatial
discretization and of the numerical scheme (Vuorinen et al., 2014; Rezaeiravesh and Liefvendahl, 2018; Breuer, 1998; Mon-
tecchia et al., 2019). Further, truncation errors corrupt the high-wavenumber range of the solution, also complicating the use
of dynamic LES closure models whereby the information from the small scales of motion is leveraged to evaluate the SGS
diffusion (Germano et al., 1991). Notwithstanding these limitations, such schemes have been heavily employed in the past in
both the geophysical and engineering flow communities, and are the de-facto standard in the wind engineering one, where most
of the numerical simulations are carried out using second-order-accurate Finite-Volume- (FV-) based solvers (Nilsson et al.,
2008; Stovall et al., 2010; Churchfield et al., 2010; Balogh et al., 2012; Churchfield et al., 2013; Shi and Yeo, 2016, 2017;
García-Sánchez et al., 2017, 2018). Note that the studies conducted with FV-based solvers are mainly focused on first- and
second-order flow statistics, which are themselves not sufficient to fully characterize turbulence– and related transport– in the
ABL.

The present study aims at bridging this knowledge gap by analyzing quality and reliability of a second-order-accurate FV
solver for the LES of ABL flow, with a lens on higher-order statistics, energy spectra, spatial autocorrelations and turbulence
topology. The analysis is carried out leveraging the OpenFOAM® framework (Weller et al., 1998; De Villiers, 2006; Jasak
et al., 2007). A suite of simulations is carried out whereby physical and numerical parameters are varied. The predictions from
the solver are contrasted against the results from the Albertson and Parlange (1999) PSFD code.

The work is organized as follows. Section 2 briefly summarizes the governing equations, the numerical methods and the
set-up of the problem, along with a description of the simulated cases and of the post-processing procedure. The results
are proposed in §3. The conclusions are drawn in §4. In the Appendix, the sensitivity of the solution to model constants,
interpolation schemes and numerical solvers is reported.





## 2 Methodology

### 2.1 Governing equations and numerical schemes

In the following, vector and index notations are used interchangeably, according to needs, in a Cartesian reference system. The spatially-filtered Navier-Stokes equations are considered,

$$\nabla \cdot \mathbf{u} = 0 \ , \tag{1}$$

$$\frac{\partial \mathbf{u}}{\partial t} + \nabla \mathbf{u} \cdot \mathbf{u} = -\frac{1}{\rho}\nabla \tilde{p} + \nabla \cdot \boldsymbol{\tau} - \nabla \cdot \boldsymbol{\tau}^{SGS,dev} - \frac{1}{\rho}\nabla P \ , \tag{2}$$

where $\mathbf{u} = (u_1, u_2, u_3)$ is the spatially-filtered velocity field along the streamwise $(x_1)$, vertical $(x_2)$ and spanwise $(x_3)$ coordinate directions, $t$ is the time, $\rho$ is the (constant) fluid density, $\tilde{p} \equiv p + \frac{1}{3}\tau_{kk}^{SGS}$ is the pressure term with an additional contribution from the Sub-Grid kinetic energy $(\frac{1}{2}\tau_{kk}^{SGS})$, $\boldsymbol{\tau}$ is the filtered viscous-stress tensor, $\boldsymbol{\tau}^{SGS,dev}$ is the deviatoric part of the SGS-stress tensor. In addition, the term $-\frac{1}{\rho}\nabla P$ is a constant pressure gradient, here assumed to be constant and uniform, responsible for driving the flow. The filtered viscous tensor is $\boldsymbol{\tau} = -2\nu\mathbf{S}$, where $\nu = \mathrm{const}$ is the kinematic viscosity of the Newtonian fluid and $\mathbf{S}$ is the resolved (in the LES sense) rate-of-strain tensor. For the SGS-stress tensor, the static Smagorinsky model is used,

$$\boldsymbol{\tau}^{SGS,dev} = -2\nu^{SGS}\mathbf{S} = -2(C_S\Delta)^2|\mathbf{S}|\mathbf{S} \ , \tag{3}$$

where $\nu^{SGS}$ is the SGS eddy viscosity, $C_S$ is the Smagorinsky coefficient (Smagorinsky, 1963), $\Delta = (\Delta x_1 \Delta x_2 \Delta x_3)^{1/3}$ is a local length-scale based on the volume of the computational cell (Scotti et al., 1993), and $|\mathbf{S}| = \sqrt{2\mathbf{S}:\mathbf{S}}$ quantifies the magnitude of the rate of strain. In the present work, $C_S = 0.1$, unless otherwise specified. The authors would like to point out that dynamic Smagorinsky models are in general preferred to the static one for the LES of ABL flows (Germano et al., 1991; Lilly, 1992; Meneveau et al., 1996; Porté-Agel, 2004; Bou-Zeid et al., 2005). Dynamic models evaluate SGS stresses via first-principles-based constraints, feature improved dissipation properties when compared to the static Smagorinsky one (especially in the vicinity of solid boundaries) and, foremost, are parameter-free. The choice made in the present study is motivated by problematics encountered when using the available dynamic Lagrangian model in preliminary tests. Note that, however, while SGS dissipation plays a crucial role in PSFD solvers, truncation errors typically overshadow SGS stress contributions in the second-order FV-based ones (Kravchenko and Moin, 1997). The static Smagorinsky SGS model used herein might hence perform similarly to dynamic SGS models for the considered flow set-up. This observation is supported by the results of Majander and Siikonen (2002).

The large-scale separation between near-surface and outer-layer energy-containing ABL motions poses stringent resolution requirements to numerical modelers if all of the energy-containing motions necessitate to be resolved. To reduce the computational cost of such simulations, the near-surface region is typically bypassed, and a phenomenological wall-layer model is leveraged instead to account for the impact of near-wall (inner-layer) dynamics on the outer-layer flow (Mason, 1994; Piomelli





and Balaras, 2002; Piomelli, 2008; Bose and Park, 2018). This approach is referred to as Wall-Modeled Large-Eddy Simula-
tion (WMLES), and is used herein. ABL flows are typically in fully-rough aerodynamic regime with the underlying surface
(Stull, 1988), hence a rough-wall wall-layer model is required to close the equations at the surface. Such a procedure is stan-
dard practice in WMLES of ABL flows (see e.g., Albertson and Parlange, 1999). In the present work, an algebraic wall-layer
equilibrium model for surfaces in fully-rough aerodynamic regime has been implemented, based on the logarithmic law of the
wall,

$$|\mathbf{u}| = \frac{u_\tau}{\kappa} \ln\left(\frac{x_2}{x_{2,0}}\right) , \tag{4}$$

where $|\mathbf{u}| \equiv \sqrt{u_1^2 + u_3^2}$ is the norm of the velocity at a certain distance from the ground level, $u_\tau$ is the friction velocity (see
Sub-Section 2.2 for details), $\kappa$ is the von Kármán constant, $x_2$ is the distance from the ground level and $x_{2,0}$ is the so-called
aerodynamic roughness length, which quantifies the drag of the underlying surface. Here, $\kappa = 0.41$ and $x_{2,0} = 0.1$ m. Specifi-
cally, the kinematic wall shear stress is assumed to be proportional to the local velocity gradient (Boussinesq approximation),

$$\tau_{\alpha 2, w} = (\nu + \nu_t) \frac{\partial u_\alpha}{\partial x_2}\bigg|_w = (\nu + \nu_t) \frac{u_\alpha}{x_2}, \quad \alpha = 1, 3 , \tag{5}$$

with $\nu_t$ being the total eddy viscosity at the wall. From the log-law (Eq. 4) evaluated at the first cell-center, one can write $u_\tau = (\kappa |\mathbf{u}|)/(\ln(x_2/x_{2,0})$. Using the definition of friction velocity $u_\tau = \sqrt{\tau_{\alpha 2, w} |\mathbf{u}|/u_\alpha}$ for $\alpha = 1, 3$ (no summation over repeated
indices) along with Eq. 5 and rearranging, the total eddy viscosity reads

$$\nu_t = \left(\frac{\kappa |\mathbf{u}|}{\ln\left(\frac{x_2}{x_{2,0}}\right)}\right)^2 \frac{x_2}{|\mathbf{u}|} - \nu , \tag{6}$$

which is the formulation implemented herein. Note that $\nu + \nu_t \approx \nu_t$ in boundary-layer flows in fully-rough aerodynamic
regime, so that $\nu$ could be neglected without loss of accuracy.

In the OpenFOAM® framework, considered in the present work, the FV method is used on a co-located grid. The integral
version of the filtered Navier-Stokes equations is solved on every control volume, leveraging the Gauss divergence theorem to
relate volume integrals to surface integrals. Unknowns are evaluated at face-centers and are assumed to be constant on each
face, yielding an overall second-order spatial accuracy (Churchfield et al., 2010). A range of interpolation schemes is available,
spanning from first-order upwind to higher-order ones. The linear interpolation scheme is considered herein, unless otherwise
stated. Simulations are carried out using the PISO fractional step method to solve the system of equations (Issa, 1985), and an
implicit Adams-Moulton time-stepping scheme is chosen for time integration (Ferziger and Peric, 2002).

## 2.2 Problem set-up

An extensive series of WMLESs of ABL flow (open-channel-flow set-up) is performed. Tests are carried out in the domain
$[0, L_1] \times [0, L_2] \times [0, L_3]$ with $L_1 = 2\pi h$, $L_2 = h$, $L_3 = \frac{4}{3}\pi h$, where $h = 1000$ m denotes the width of the open channel. The





**Table 1.** Tabulated list of cases.

| label | C-$2\pi$ | B-$4\pi$ | B-$2\pi$ | B-$\pi$ | F-$2\pi$ | R-$2\pi$ |
|---|---|---|---|---|---|---|
| grid resolution | $32 \times 32 \times 32$ | $32 \times 64 \times 32$ | $64 \times 64 \times 64$ | $128 \times 64 \times 128$ | $128 \times 128 \times 128$ | $64 \times 64 \times 64$ |
| aspect ratio | $2\pi$ | $4\pi$ | $2\pi$ | $\pi$ | $2\pi$ | $2\pi$ |
| symbol | dashed line | dash-dotted line | circles | dotted line | solid line | full circles |

symmetric boundary condition is imposed on the top of the computational domain, no-slip applies at the lower surface, and periodic boundary conditions are enforced along each side. A pressure gradient term $-\frac{1}{\rho}\partial P/\partial x_1 = 1$ m/s$^2$ drives the flow along the $x_1$ coordinate direction, yielding $u_\tau = 1$ m/s. The kinematic viscosity is set to $10^{-7}$ m$^2$/s in the bulk of the flow, resulting in $Re_\tau = 10^7$.

Five cases are run, spanning different grid resolutions and aspect ratios. The mesh is Cartesian, with a uniform stencil along each direction. In the following, $N_i$ denotes the number of cell-centers along the $i$-th direction. The baseline calculation B-$2\pi$ is performed over $64^3$ control volumes. Two cases with the same aspect ratio $\Delta x_1/\Delta x_2 = 2\pi$ are run– the simulation C-$2\pi$ over a coarser grid ($32^3$ control volumes) and the simulation F-$2\pi$ over a finer grid ($128^3$ control volumes). Two additional cases are considered, with the same number of grid points along the vertical direction as in B-$2\pi$ and different aspect ratios– the simulation B-$4\pi$, with aspect ratio $\Delta x_1/\Delta x_2 = 4\pi$ ($N_1 \times N_2 \times N_3 = 32 \times 64 \times 32$), and the simulation B-$\pi$, with aspect ratio $\Delta x_1/\Delta x_2 = \pi$ ($N_1 \times N_2 \times N_3 = 128 \times 64 \times 128$). Note that the grid-aspect-ratio sensitivity analysis is carried out by refining the grid only along the horizontal directions, in line with the approach of Park and Moin (2016). Preliminary tests indeed showed that, for the given resolution, ABL flow statistics are more sensitive to variations in the horizontal grid stencil and aspect ratio than in the vertical ones. The chosen grid resolutions are in line with those typically used in studies of ABL flows (see, e.g., Salesky et al., 2017). All the calculations satisfy the Courant-Friedrichs-Lewy (CFL) condition $Co \lesssim 0.1$, where $Co$ is the Courant number. Runs are initialized from a fully-developed open-channel-flow simulation at equilibrium, and time integration is carried out for 100 eddy turnover times, where the eddy turnover time is defined as $h/u_\tau$. Flow statistics are the result of an averaging procedure in the horizontal plane of statistical homogeneity of turbulence ($x_1 x_3$) and in time over the last 60 eddy turnover times. The procedure yields well converged statistics throughout the considered cases. In the following, the space/time averaging operation is denoted by $\langle \cdot \rangle$.

Results in the present study are contrasted against corresponding ones from the Albertson and Parlange (1999) mixed PSFD code. Said code is based on an explicit second-order-accurate Adams-Bashforth scheme for time integration and on a fractional-step method for solving the system of equations. A single run, the reference simulation R-$2\pi$, was carried out with the PSFD solver at a resolution of $64^3$ co-location nodes, using a static Smagorinsky SGS model with $C_s = 0.1$, a rough wall-layer model with $x_{2,0} = 0.1$ m, $Co \lesssim 0.1$, and the same initialization and averaging procedure as the one considered for the FV runs. A summary of the simulated cases is given in Tab. 1.


## 3  Results

This Section is devoted to the analysis of velocity statistics, spectra and autocorrelations from the second-order FV-based solver, along with detailed considerations on turbulence topology and momentum transfer mechanisms. Mean streamwise velocity, resolved Reynolds stresses and higher-order statistics are discussed in §3.1. Sub-Section 3.2 focuses on velocity spectra and spatial autocorrelations, and a discussion on the turbulence topology based on conditionally-averaged flow field and quadrant analysis can be found in §3.3.

### 3.1  Mean profiles

In Fig. 1, the vertical structure of the normalized mean streamwise velocity ($\langle u_1 \rangle^+$) and resolved shear Reynolds stress ($-\langle u_1' u_2' \rangle^+$) is shown for all of the considered cases. The mean streamwise velocity at the first two cell-centers off the wall is consistently underpredicted, whereas a positive Log-Layer Mismatch (LLM) is observed in the bulk of the flow (Kawai and Larsson, 2012). This behavior could have been anticipated, as the wall shear stress is evaluated using the instantaneous horizontal velocity at the first cell-center off the wall. A number of procedures has been proposed to alleviate the LLM, including modifying the SGS-stress model in the near-wall region (Sullivan et al., 1994; Porté-Agel et al., 2000; Chow et al., 2005; Wu and Meyers, 2013), shifting the matching location further away from the wall (Kawai and Larsson, 2012), and carrying out a local horizontal/temporal filtering operation (Bou-Zeid et al., 2005; Xiang et al., 2017). In preliminary runs, the authors applied the same approach as in Kawai and Larsson (2012) to mitigate the LLM, but observed an enhanced sensitivity of mean velocity profiles to grid resolution and matching location that suggested that alternative procedures might work better for the considered solver. The results herein proposed are hence representative of the OpenFOAM® solver with the standard wall-layer treatment— the set-up that is most commonly adopted when using this code (see, for instance, Churchfield et al. (2010); Shi and Yeo (2017)). Note that a positive LLM is observed even when using the PSFD solver, in spite of a spatial low-pass filtering operation that is carried out on the horizontal velocity before the evaluation of the surface shear stress (Bou-Zeid et al., 2005). The one in Fig. 1(a) is indeed the expected mean velocity profile for PSFD solvers coupled with the static Smagorinsky model (see Meneveau et al., 1996; Bou-Zeid et al., 2005) and advocates for the use of alternative strategies to overcome the LLM therein as well. In Fig. 1(b), resolved Reynolds stresses are compared to the theoretical profile of the total stress $\tau_{12}^{tot} = (-\langle u_1' u_2' \rangle + \tau_{12}^{SGS} + \tau_{12}) = u_\tau^2 (1 - x_2/h)$. The profiles from the FV-based solver feature a strong sensitivity to grid resolution and aspect ratio, and start off relatively slow from the wall when compared to those from the PSFD one, throughout all the considered cases. For instance, at $x_2/h \approx 0.01$, the resolved Reynolds stresses from the R-$2\pi$ case account for $21\%$ of the total shear stress, whereas they account for only $2\%$ in the corresponding B-$2\pi$ case, $6\%$ in the B-$\pi$ case, and $8\%$ in the F-$2\pi$ one. This behavior is likely due to truncation errors that damp the energy of high-wavenumber momentum-carrying modes in the near-surface region (see discussion in §3.2), which controls in large part the overall solution (Van Driest, 1956; Kawai and Larsson, 2012). The present results suggest that the impact of the SGS model on the global solution might be larger for FV-based solvers than for PSFD-based ones via SGS near-wall effects. This conclusion, however, is at odds with some of the numerical experiments that were conducted, where the solution was found to be poorly sensitive– when compared to the

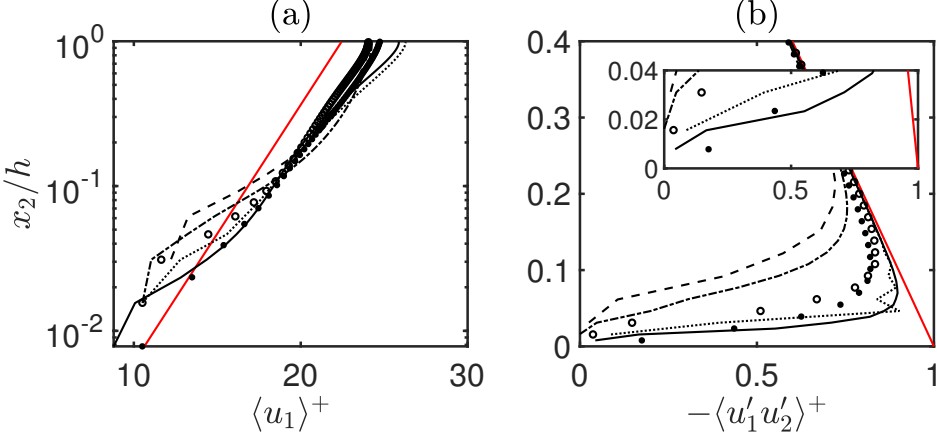

**Figure 1.** Vertical structure of mean streamwise velocity $\langle u_1 \rangle^+ = \langle u_1 \rangle / u_\tau$ (a) and resolved stress $\langle u_1' u_2' \rangle^+ = \langle u_1' u_2' \rangle / u_\tau^2$ (b). Red lines denote the phenomenological logarithmic-layer profile (a) and the theoretical profile for the total Reynolds stress (b). The other lines and symbols are defined in Table 1.

one from the PSFD-based-solver– to details of the near-wall numerical procedure (e.g., using or not a wall-damping function). Truncation errors might again be responsible for said behavior.

Turbulence intensities are shown in Fig. 2, where $(\cdot)'$ denotes the Root Mean Square (RMS) of the fluctuations and $(\cdot)'^+ \equiv (\cdot)'/u_\tau$. The profiles are extremely sensitive to the grid resolution in the horizontal coordinate directions and start off relatively

slow at the wall when compared to the R-$2\pi$ case and to the reference profile from Hultmark et al. (2013). As a result, the velocity fluctuations are consistently underpredicted in the very near-wall region ($x_2/h \leq 0.025$). On the contrary, the $u_1'^+$-peak values are overpredicted, whereas the $u_2'^+$- and $u_3'^+$- peak values are underpredicted, except for the finest horizontal-grid-resolution runs (cases B-$\pi$ and F-$2\pi$). The overshoot in the peak of $u_1'^+$ and the underestimation of $u_2'^+$ and $u_3'^+$ in the surface-layer region are a well-known problem of FV-based WMLES (Bae et al., 2018). Lack of energy redistribution

via pressure fluctuation from shear-generated $u_1'^+$ to $u_2'^+$ and $u_3'^+$ is the root cause of said behavior, and possible mitigation strategies include allowing for wall transpiration (Bose and Moin, 2014; Bae et al., 2018). Grid refinement in the horizontal directions leads to an improved matching between the FV and the PSFD solver, both in terms of shape and magnitude.

Skewness and kurtosis of the streamwise velocity ($S_1$ and $K_1$, respectively) are shown in Fig. 3, along with the transfer efficiency coefficient, $r_{12} = -\langle u_1' u_2' \rangle / (u_1' u_2')$. Average values of said flow statistics in the surface layer are shown in Tab.

2, where spurious near-wall effects are neglected by constraining the averaging to the interval $0.2 \leq x_2/h \leq 0.4$. Recall that the constancy of $S_1 \approx -0.3$, $K_1 \approx 3$, and $r_{12} \approx 0.3$ in the surface layer of the ABL is a manifestation of the self-similar nature of ABL turbulence therein (Del Álamo et al., 2006). Both the PSFD- and FV-based solvers predict a spurious maximum $S_1 \approx 1$ at the first node off the wall, followed by a monotonic decrease in the $x_2/h \lesssim 0.2$ range. The observed near-surface maximum may be originated from wall-blocking effects (Perot and Moin, 1995; Bae et al., 2018) such as splats, local regions

of stagnation flow resulting from fluid impinging on a wall, investigated in Perot and Moin (1995). Note that near-wall effects



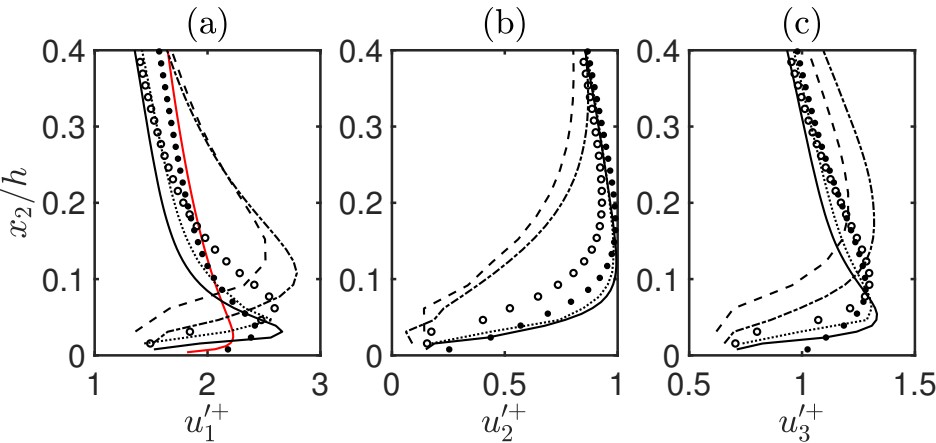

**Figure 2.** Vertical structure of streamwise velocity RMS (a), vertical velocity RMS (b) and spanwise velocity RMS (c). Lines and symbols are defined in Table 1. The red line denotes the reference profile from Hultmark et al. (2013).

**Table 2.** Average values of skewness, kurtosis and transfer efficiency coefficient in the interval $0.2 \leq x_2/h \leq 0.4$.

| simulation | C-$2\pi$ | B-$4\pi$ | B-$2\pi$ | B-$\pi$ | F-$2\pi$ | R-$2\pi$ |
|---|---|---|---|---|---|---|
| $\langle S_1 \rangle_{surface\,layer}$ | $-0.0660$ | $-0.1566$ | $-0.0013$ | $0.1092$ | $0.1509$ | $-0.3572$ |
| $\langle K_1 \rangle_{surface\,layer}$ | $3.1362$ | $3.3078$ | $3.2864$ | $3.1223$ | $3.2434$ | $2.7480$ |
| $\langle r_{12} \rangle_{surface\,layer}$ | $0.4422$ | $0.4169$ | $0.4964$ | $0.4867$ | $0.5157$ | $0.4331$ |

extend deeper within the boundary layer for the FV-based runs and, further, the profiles remain positive throughout, except for the two coarse-resolution cases (C-$2\pi$ and B-$4\pi$). Grid refinement in the horizontal directions improves the matching between the FV-based and the PSFD-based solvers in the near-wall region, and accelerates the convergence of the profiles to the constant surface-layer values. $K_1$ profiles also feature a spurious maximum at the wall, and approximately constant values are reached

above $x_2/h \approx 0.2$ for the B-$2\pi$, B-$\pi$, and F-$2\pi$ cases, as well as for the R-$2\pi$ case. On the contrary, no constant-$K_1$ layers are observed for the C-$2\pi$ and B-$4\pi$ cases. The constant-$K_1$ value is consistently overpredicted, signaling a flow field that is populated by a number of rare events larger than the one in real-world neutrally-stratified ABLs. From the transfer efficiency profiles shown in Fig. 3(c) it is also apparent that both PSFD- and FV-based solvers predict a flow field populated by coherent structures that are more efficient in transferring momentum than those in measured ABLs (Bradshaw, 1967). The profiles from

the FV-based solver reach an approximately constant $r_{12}$ value further aloft ($x_2/h \approx 0.2$) when compared to the reference simulation R-$2\pi$.

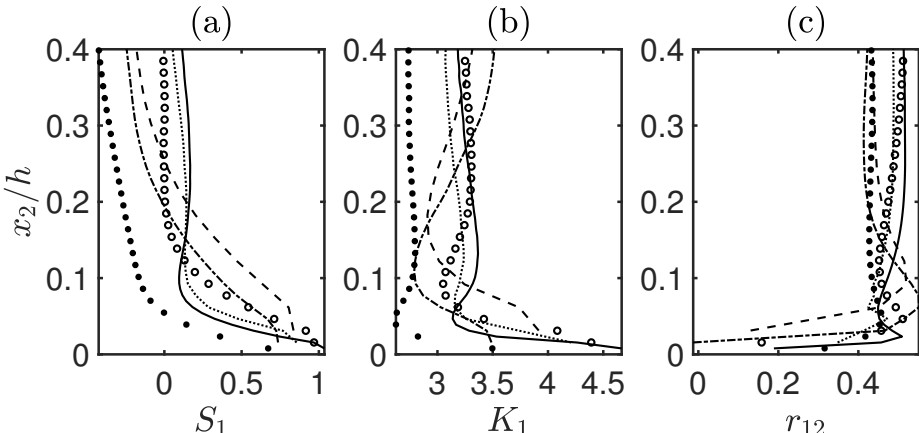

**Figure 3.** Vertical structure of skewness of streamwise velocity (a), kurtosis of streamwise velocity (b) and transfer efficiency coefficient (c). Lines and symbols are defined in Table 1.

### 3.2 Spectra and autocorrelations

In this Sub-Section, spectra and spatial autocorrelations of the streamwise velocity fluctuations are analyzed, to quantify the distribution of energy density across scales and the spatial coherence of the simulated ABLs.

The one-dimensional spectrum of the streamwise velocity fluctuations ($E_{11}$) is featured in Fig. 4(a) for all of the considered cases. The profiles are contrasted against the phenomenological production-range and inertial-sub-range power-law profiles ($k^{-1}$ and $k^{-5/3}$, respectively). In the production range, the spectra are sensitive to the horizontal grid resolution, with an apparent decrease in the power-law exponent as the resolution is increased. The profiles from the simulations C-$2\pi$ and B-$4\pi$ and those from the simulations B-$\pi$ and F-$2\pi$ are similar, highlighting once again that the solution is more sensitive to the horizontal grid resolution than to the vertical one, and that the aspect ratio does not play an important role herein. In the high-wavenumber range, the profiles feature a rapid decay of energy density, regardless of the resolution or the aspect ratio, and the decay is shifted towards higher wavenumber as the horizontal grid resolution is increased. Cases C-$2\pi$ and B-$4\pi$ also display an unphysical pile-up of energy near the cut-off frequency. It is evident that inertial-range turbulence dynamics may not be well captured in the simulated cases, and this fact might complicate the use of dynamic procedures based on the Germano et al. (1991) identity. The results suggest that, for the considered resolutions, neither grid refinement nor the reduction of the aspect ratio help circumvent this limitation (no trend is observed). Note, however, that the contribution of the inertial-sub-range portion of the spectrum to the overall energy is modest, ranging from $10\%$ to $15\%$ for all the simulated cases (see Tab. 3). On the contrary, predictions from the PSFD-based solver are not sensitive to grid resolution (not shown), and feature a very good agreement with the phenomenological $-5/3$ power-law profile in the inertial sub-range. A further characterization of the energy dynamics is given in Fig. 4(b), where the premultiplied spectrum $k_1 h E_{11}/u_\tau^2$ is considered at selected heights for the cases F-$2\pi$ and R-$2\pi$. Premultiplied spectra profiles provide information on the coherence of the flow and in particular

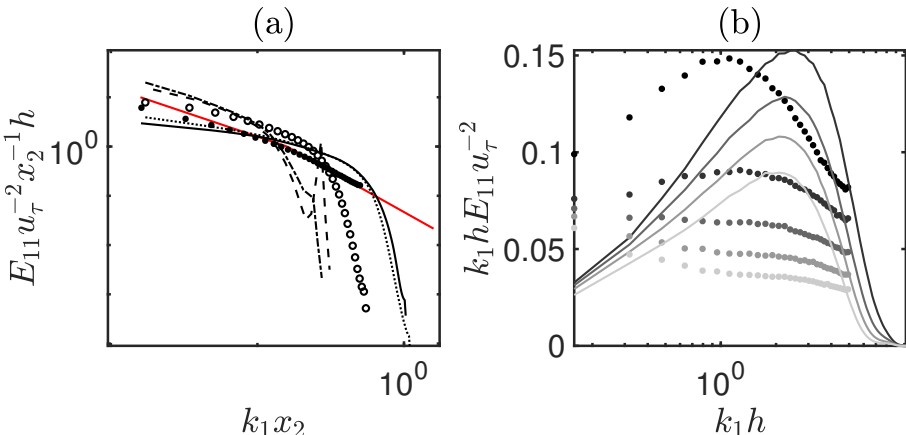

**Figure 4.** (a) Normalized one-dimensional spectrum of streamwise velocity at height $x_2/h \approx 0.1$. Lines and symbols: solid red line, $(k_1 x_2)^{-1}$ in the production range and $(k_1 x_2)^{-5/3}$ in the inertial sub-range; for all the other lines and symbols please refer to Table 1. (b) Premultiplied one-dimensional spectrum of streamwise velocity from the simulations R-2$\pi$ (full circles) and F-2$\pi$ (solid lines). Dark to light gray lines correspond to heights from $x_2/h \approx 0.1$ to $x_2/h \approx 0.5$.

on the so-called Large- and Very-Large- Scale Motions (LSMs and VSLMs, respectively). LSMs consist of single hairpin packets whose legs form counter-rotating rolls generating a low-velocity streamwise-elongated streak, also inducing high-momentum bulges on the sides of said streak. Velocity correlation analyses have shown that LSMs typically extend up to $3h$ in

the streamwise direction and $h$ in the spanwise direction. VLSMs arise due to clustering of such structures in the streamwise direction, and can reach streamwise extents of over $20h$ in boundary-layer flows. These structures are responsible for carrying more than half of the kinetic energy and Reynolds shear stress and are a persistent feature of the surface and outer layers of both aerodynamically smooth and rough walls (Hutchins and Marusic, 2007a; Fang and Porté-Agel, 2015). Numerous works have recently been devoted to the identification and characterization of LSMs and VLSMs in wall-bounded flows, both from a

numerical and experimental perspective (Kim and Adrian, 1999; Balakumar and Adrian, 2007; Monty et al., 2007; Hutchins and Marusic, 2007b; Fang and Porté-Agel, 2015). The current domain, of modest dimensions, can accommodate only LSMs (Lozano-Durán and Jiménez, 2014), commonly identified in premultiplied spectra by a local maximum at the streamwise wavenumber $k_1/h \approx 1$. As apparent from Fig. 4(b), the premultiplied spectrum from the FV-based solver underpredicts the streamwise extent of LSMs, with a maximum located at $k_1/h \approx 3$. The PSFD-based solver, on the contrary, succeeds in

capturing LSMs, despite the modest extent of the computational domain.

To gain insight on the spatial coherence of the flow field, contour lines of the two-dimensional autocorrelation of the streamwise velocity ($R_{11}^{2D}$) in the $x_1 x_3$ plane are shown in Fig. 5. Contours from the F-2$\pi$ case (Fig. 5(b)) are representative of a flow field less correlated along both streamwise and spanwise directions than the one from the R-2$\pi$ case (Fig. 5(a)), and also more isotropic (note that the scales in Fig. 5(b) differ from those in Fig. 5(a)). For example, the ellipse-shaped contour line at level

0.3 from the R-2$\pi$ simulation is characterized by eccentricity $e \approx 0.9965$, while the corresponding value for the F-2$\pi$ simula-



**Table 3.** Ratio of inertial sub-range energy ($E_{inertial}$) to total energy ($E_{total}$) at $x_2/h \approx 0.1$. The the total energy is computed as the integral of the normalized spectrum across the whole available wavenumber range, whereas the inertial-range energy is obtained by integration in the wavenumber region with slope $-5/3$ or steeper.

| simulation | C-$2\pi$ | B-$4\pi$ | B-$2\pi$ | B-$\pi$ | F-$2\pi$ | R-$2\pi$ |
|---|---|---|---|---|---|---|
| $E_{inertial}/E_{total}$ | 0.1380 | 0.0951 | 0.1338 | 0.0947 | 0.1564 | 0.1188 |

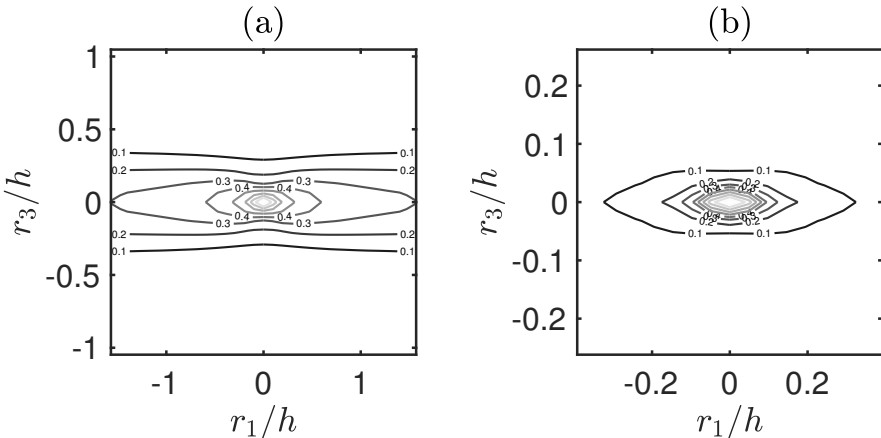

**Figure 5.** Contours of two-dimensional spatial autocorrelation of streamwise velocity at height $x_2/h \approx 0.1$, from the simulation R-$2\pi$ (a) and from the simulation F-$2\pi$ (b). Dark to light gray lines correspond to contour levels from $0.1$ to $0.9$ with increments of $0.1$.

tion is $e \approx 0.9473$. Note that the quality of the computed flow statistics is not impacted by the fact that the current domain size prevents some of the contour lines in the R-$2\pi$ case from closing (Lozano-Durán and Jiménez, 2014).

The one-dimensional autocorrelation function ($R_{11}$), shown in Fig. 6 along the streamwise and spanwise coordinate directions for all of the considered cases, further supports the above statements. From the profiles from the R-$2\pi$ simulation it is

clear that the extension of the selected domain is not suffiecient to capture all the dynamics, as $R_{11}$ remains finite in the available $r_1/h$ range. Along the spanwise direction, $R_{11}$ features the expected negative lobes, which highlight the presence of high- and low-momentum streamwise-elongated streaks flanking each others in the streamwise direction, in line with findings from previous studies focused on the coherence of wall-bounded turbulence. Throughout the considered FV-based solver cases, $R_{11}$ decays very rapidly along the streamwise and spanwise directions, more so as the grid is refined. Further, the negative lobes in

the spanwise autocorrelation weaken in the B-$\pi$ and F-$2\pi$ cases, and spread over a much larger separation distance. A quantitative measure of the coherence length of the flow is provided in Tab. 4, where the integral lengths $\Lambda_{r_1, u_1}$ and $\Lambda_{r_3, u_1}$ are reported, in a comparison with corresponding values from direct numerical simulations of the channel flow at $Re_\tau = 2000$ from Sillero et al. (2014). The integral length $\Lambda_{r_i, u_1}$ is obtained by integration of the $R_{11}$ function along the $i$-th direction, from $r_i = 0$ to the first zero (if any) or to the closest intersection with $R_{11} = 0.05$, in line with the procedure outlined in Sillero et al. (2014).



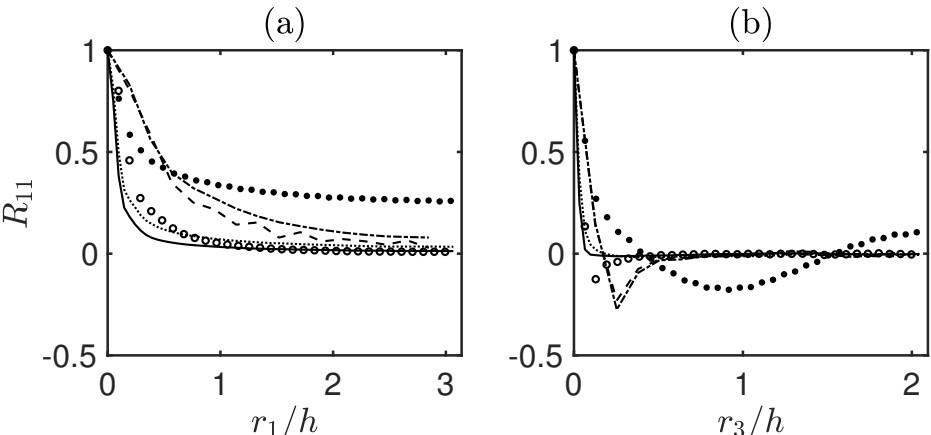

**Figure 6.** One-dimensional spatial autocorrelation of streamwise velocity at height $x_2/h \approx 0.1$, along the streamwise direction (a) and along the spanwise direction (b). Lines and symbols are defined in Table 1.

**Table 4.** Integral lengths at height $x_2/h \approx 0.15$.

| simulation | C-$2\pi$ | B-$4\pi$ | B-$2\pi$ | B-$\pi$ | F-$2\pi$ | R-$2\pi$ | Sillero et al. (2014) |
|---|---|---|---|---|---|---|---|
| $\Lambda_{r_1,u_1}/h$ | 0.5455 | 0.6496 | 0.2320 | 0.2637 | 0.1203 | 1.2810 | 2.1440 |
| $\Lambda_{r_3,u_1}/h$ | 0.0709 | 0.0708 | 0.0379 | 0.0415 | 0.0293 | 0.1436 | 0.2021 |

While $\Lambda_{r_1,u_1}$ might be not meaningful for the R-$2\pi$ case, owing to the lack of a zero crossing of the autocorrelation function, it is apparent that the values of the coherence lengths from the FV-based solver are much smaller than expected, and that the grid refinement procedure leads to a further reduction of both $\Lambda_{r_1,u_1}$ and $\Lambda_{r_3,u_1}$. These findings highlight a flow field that is less correlated than realistic ABL flows, thus suggesting that the FV-based solver may not be capable of representing coherent structures– and associated momentum-transfer mechanisms– of ABL turbulence.

**3.3   Instantaneous horizontal contours**

To further substantiate the lack of coherence in the FV flow fields, horizontal instantaneous snapshots of normalized streamwise velocity fluctuations are shown in Fig. 7 for the simulations R-$2\pi$ and F-$2\pi$. The normalized velocity fluctuation is defined as $(u_1 - \langle u_1 \rangle_{x_1 x_3})/u_1''$, where averages are carried out in space over the selected horizontal plane. Streamwise-elongated bulges of uniform high and low momentum are apparent in the R-$2\pi$ flow field (Fig. 7(a)). These are the typical flow patterns encountered

in boundary-layer flows and have been the object of significant studies in both geophysics and engineering (Balakumar and Adrian, 2007; Hutchins and Marusic, 2007a; Fang and Porté-Agel, 2015). The streamwise velocity field from the FV-based solver (Fig. 7(b)) exhibits a less coherent flow field when compared to the one from the PSFD-based solver. Differences are particularly stark in the spanwise direction, where thin structures populate the boundary layer and LSMs are not clearly

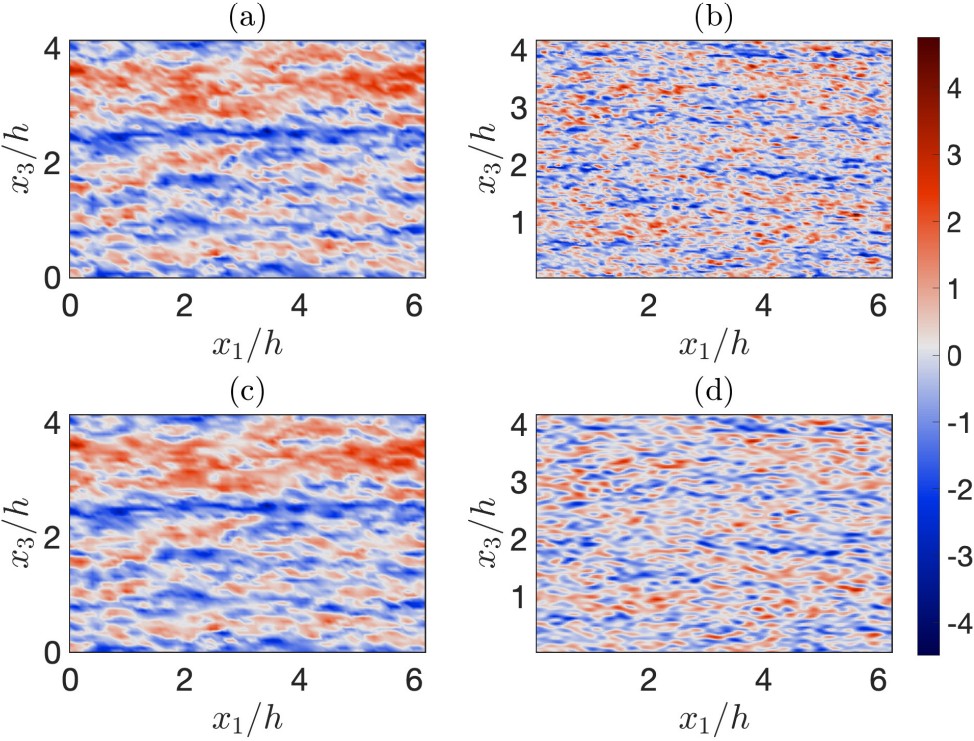

**Figure 7.** Instantaneous snapshots of normalized streamwise velocity fluctuations at $x_2/h \approx 0.1$ from simulations R-$2\pi$ (a), F-$2\pi$ (b), R-$2\pi$ filtered (c) and F-$2\pi$ filtered (d). The normalized velocity fluctuation is defined as $(u_1 - \langle u_1 \rangle_{x_1 x_3})/u_1''$, where averages (and fluctuations therefrom) are evaluated in space over the selected horizontal plane. A low-pass spatial filtering operation was carried out to obtain the flow field in panels (c) and (d), using a sharp-spectral-cut-off kernel with support $\ell_1/h \times \ell_3/h = 3x_2/h \times x_2/h$– approximately the extent of LSMs in the ABL.

detectable. To gain further insight on the problem, in the spirit of LES, the instantaneous velocity snapshots have been spatially

low-pass filtered using a sharp-spectral-cut-off kernel with support $\ell_1/h \times \ell_3/h = 3x_2/h \times x_2/h$– approximately the extent of LSMs across the ABL (see Fig. 7(c) and (d)). From the filtered flow field it is indeed apparent that larger-scale patterns are present in the OpenFOAM® solution, but these are less coherent than the corresponding ones from the PSFD-based solver, and energetically weaker, thus not bringing significant contributions to autocorrelation maps.

To elucidate the mechanisms responsible the for momentum transport in the flow, the conditionally-averaged flow field

is analyzed, following the approach of Fang and Porté-Agel (2015). In Fig. 8, a visualization of the conditionally-averaged flow field is provided– the conditional event being a positive streamwise velocity fluctuation $u_1''/u_\tau$ at $r_1/h = 0$, $x_2/h = 0.5$, $r_3/h = 0$. The flow structure in the equilibrium surface layer is expected to exhibit rolls in the vertical-spanwise plane, each roll flanked by a low- and a high-momentum streamwise-elongated streaks. The roll leads to sweep and ejection pairs, which occur in correspondence of the high- and low-momentum streak respectively, and are the dominant mechanism responsible for



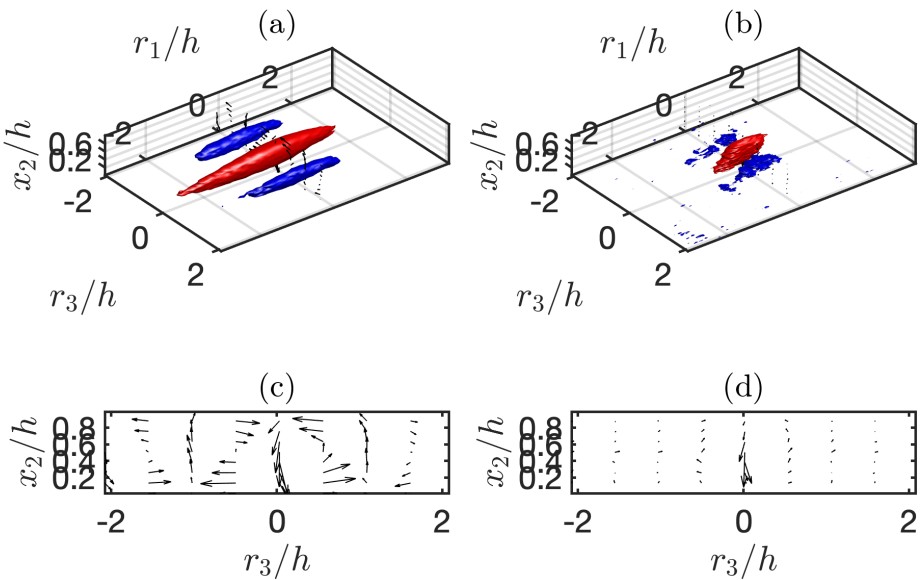

**Figure 8.** Conditionally-averaged flow field from the simulation R-$2\pi$ ((a) and (c)) and from the simulation F-$2\pi$ ((b) and (d)). The conditional event is a positive streamwise velocity fluctuation $u_1''/u_\tau$ at $r_1/h = 0$, $x_2/h = 0.5$, $r_3/h = 0$. Top: red iso-surfaces show $u_1''/u_\tau > 0.68$ (a) and $u_1''/u_\tau > 0.09$ (b); blue iso-surfaces show $u_1''/u_\tau < -0.54$ (a) and $u_1''/u_\tau < -0.04$ (b); vector fields in the spanwise-vertical planes are visualized at $r_1/h = -L_1/(4h), 0, L_1/(4h)$. Bottom: vector field in the spanwise-vertical plane at $r_1/h = 0$.

tangential Reynolds stress (Ganapathisubramani et al., 2003; Lozano-Durán et al., 694). The results from the simulation R-$2\pi$ clearly capture said mechanism, with sweeps and ejections of the same order of magnitude. A qualitatively similar pattern can be obtained from the F-$2\pi$ case, but streaks are significantly weaker when compared to those in the R-$2\pi$ case (see details in caption of Fig. 8). When the threshold is fixed to be the same as for the simulation R-$2\pi$, only positive-fluctuation patterns can be visualized, and the opposite occurs if the conditional event is a negative streamwise velocity fluctuation, signaling a

flow field where a strong sweep (ejection) contributing to the tangential Reynolds stress does not have a corresponding ejection (sweep) pattern.

To gain further insight on relative contributions of sweeps and ejection to the overall Reynolds stress, a quadrant-hole analysis is proposed hereafter (Lu and Willmarth, 1973). This technique is based on the decomposition of the velocity fluctuations into four quadrants: the first and third quadrants, "outward interactions" ($u' > 0$, $v' > 0$) and "inward interactions" ($u' < 0$,

$v' < 0$) respectively, are negative contributions to the momentum flux, whereas the second and fourth quadrants, a.k.a. "ejections" ($u' < 0$, $v' > 0$) and "sweeps" ($u' > 0$, $v' < 0$), represent positive contributions. The notation is the same as in Yue et al. (2007a), where $H$ is the hole size, $S_{i,H}$ is the Reynolds shear stress contribution to the $i$-th quadrant at hole size $H$, and $S_{i,H}^f$ is the correspondent quadrant fraction.



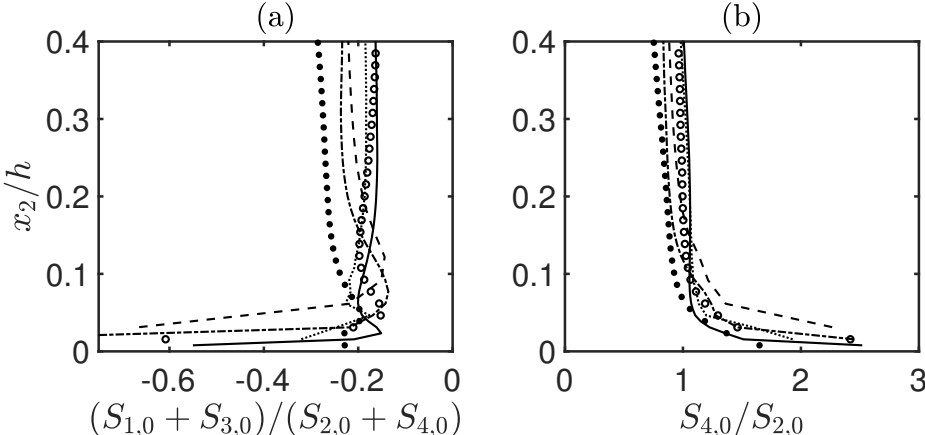

**Figure 9.** Vertical structure of event ratios: (a) ratio of negative to positive contributions to the momentum flux; (b) ratio of sweeps to ejections. Lines and symbols are defined in Table 1.

Figure 9(a) shows the exuberance ratio, defined as the ratio of negative to positive contributions to the momentum flux, $(S_{1,0} + S_{3,0})/(S_{2,0} + S_{4,0})$ (Shaw et al., 1983). The magnitude of the profile from the R-$2\pi$ simulation is larger than those from the FV runs, highlighting that outward and inward interactions have a relative contribution to the resolved Reynolds stress that is more significant for the PSFD-based solver, whereas the FV results are characterized by relatively stronger ejections and sweeps. More interestingly, from Fig. 9(b) it is apparent that the FV solver tends to favor sweeps over ejection as the mechanisms for momentum transfer in the surface layer, which is at odds with the R-$2\pi$ predictions and with findings from measurements of surface-layer flow over rough surfaces, whereby ejections are identified as the dominant momentum transport mechanism (Raupach et al., 1991). Grid refinement over the considered resolutions does not mitigate this shortcoming.

Consistently with these findings, the joint probability density function of the streamwise and vertical velocity fluctuations for the simulation F-$2\pi$ exhibits a narrower range of inner-outer interactions, as displayed in Fig. 10(b). It is also apparent that the PSFD-based solver features a larger variance, highlighting that stronger sweeps and ejections are favored when compared to those from the FV-based solver.

These observations are further supported by Fig. 11, where stress fractions are reported for values of the hole size $H$ ranging from 0 to 8. Note that larger hole sizes corresponds to contributions from more extreme events to the resolved Reynolds shear stress. Clearly, the FV-based solver severely underpredicts ejections, outward interactions and inward interactions (Fig. 11(a), (b) and (c), respectively), and slightly overpredicts extreme sweeps (Fig. 11(d) at sufficiently large hole size $H$). This mismatch is particularly apparent for the low grid-resolution cases, with a general improvement as the grid is refined. Ejections in the ABL are known to be relatively violent events, concentrated over a very thin region in the spanwise direction (Fang and Porté-Agel, 2015). The findings from Fig. 11 suggest that, at the considered grid resolutions, the FV solver is not able to correctly capture said strong local events, leading to a less coherent flow field and, possibly, to many of the observed discrepancies with the R-$2\pi$ case and with canonical ABL flow statistics.



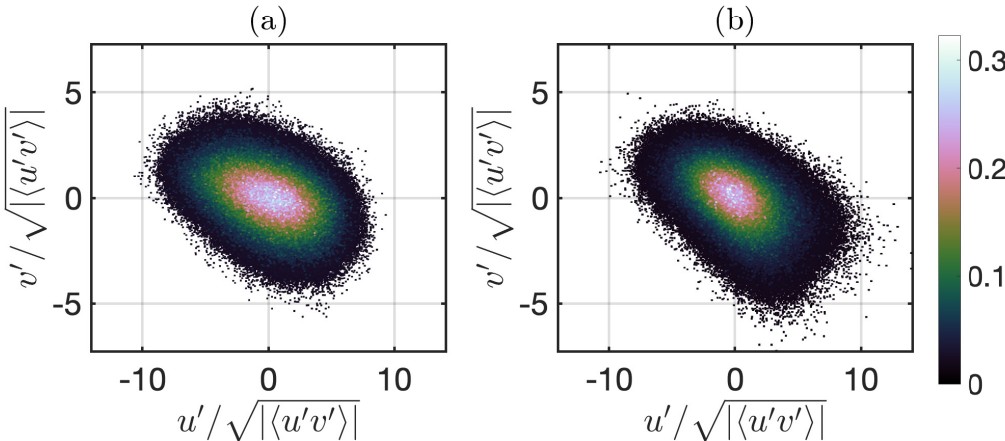

**Figure 10.** Joint probability density function of the streamwise and vertical velocity fluctuations, normalized by the resolved Reynolds shear stress, at $x_2/h \approx 0.1$. Results from the simulation R-$2\pi$(a) and from the simulation F-$2\pi$ (b).

## 4   Conclusion

The objective of the present study was to determine whether second-order-accurate FV-based solvers are suitable for WMLESs of ABL flows. A suite of simulations has been carried out using a general-purpose co-located FV solver based on second-order centered schemes within the OpenFOAM® framework, varying parameters such as grid resolution and aspect ratio. Results have been contrasted against those from a validated PSFD-based solver.

Mean velocity and resolved Reynolds stresses are found to be particularly sensitive to variations in the surface-parallel grid resolution, and a relatively good convergence to corresponding profiles from the PSFD-based solver has been observed as the grid is refined. On the contrary, higher-order velocity statistics, spectra and spatial autocorrelations are severely mispredicted across grid resolutions. Skewness, kurtosis, and transfer efficiency coefficient are not constant in the surface layer (i.e., the flow is not self-similar) and are consistently overpredicted therein. Streamwise velocity spectra exhibit no phenomenological production range, are very sensitive to variations in the grid resolution and aspect ratio, and decay too rapidly in the inertial sub-range as a result of truncation errors. Further, the spectral peaks in the premultiplied streamwise velocity spectra are shifted to higher wavenumber when compared to the reference PSFD solution, and the corresponding spatial autocorrelation of the streamwise velocity rapidly decays across the ABL along both the streamwise and spanwise coordinate directions. Consistently with these findings, instantaneous snapshots of the streamwise velocity fluctuations reveal that the flow is populated by shorter and thinner structures. The dominant mechanism supporting the tangential Reynolds stress in ABL flow – spanwise-paired sweeps and ejections– is found to be much weaker than what commonly observed in the ABL, with sweeps dominating over ejections in the surface layer, which is at odds with available measurements and with corresponding results from the PSFD-based solver. A quadrant-hole analysis highlighted how the considered FV-based solver severely underpredicts ejection events, which are notoriously localized in the spanwise direction, as well as inner and outer interactions. In the authors' opinion, this



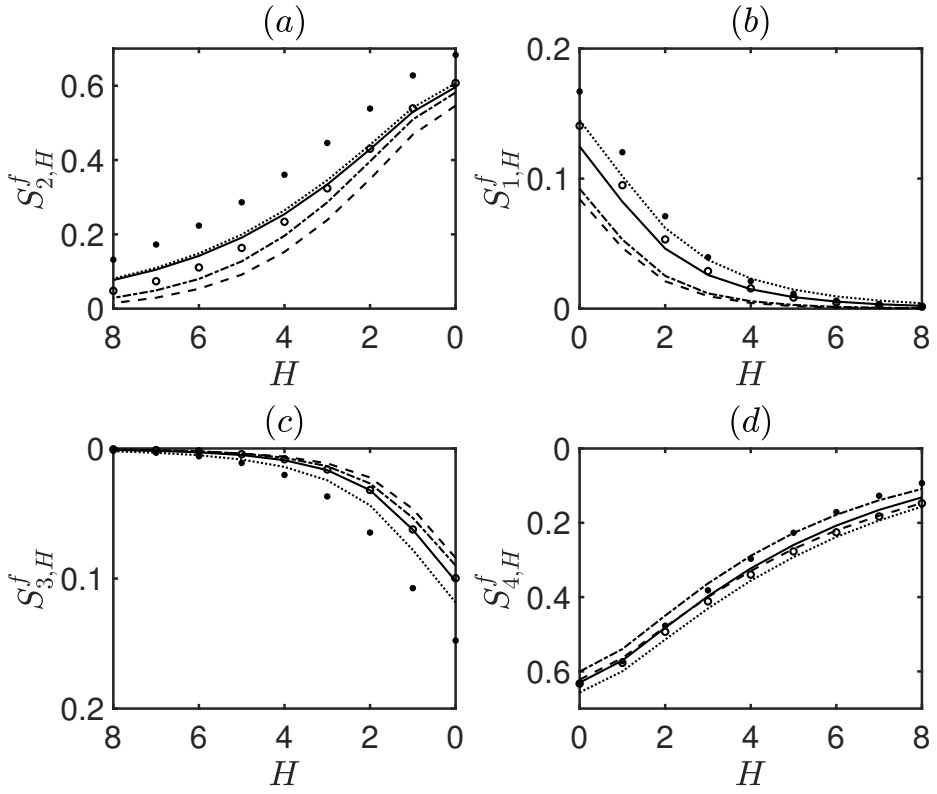

**Figure 11.** Stress fractions at $x_2/h \approx 0.1$. Profiles are normalized so that the sum of the stress fractions for $H = 0$ is unity across the cases. Lines and symbols are defined in Table 1.

underprediction of ejection events is the root-cause of many of the observed mismatches and sensitivity to grid resolution of flow statistics. This statement is partly supported by the strong sensitivity of quadrant profiles to the grid stencil, and to the approximately monotonic convergence of ejection ($S_{2,H}^f$) profiles towards the reference PSFD ones as the grid is refined.

Overall, the present findings show that truncation errors have an overwhelming impact on the predictive capabilities of second-order-accurate FV-based solvers that rely on a co-located grid set-up and centered schemes for the WMLES of ABL

flow. Although first- and second-order statistics can be considered acceptable provided sufficient grid resolution, the predictive capabilities of said solvers are relatively poor for higher-order statistics, velocity spectra, and turbulence topology.

*Code availability.* OpenFOAM® is an open-source computational-fluid-dynamics toolbox. The present study features the OpenFOAM® version 6.0, available for download at https://openfoam.org/version/6/. The Matlab scripts used for the post-processing are accessible from the GitLab repository https://gitlab.com/turbulence-columbia/miscellaneous/fv-solvers-abl-flow.



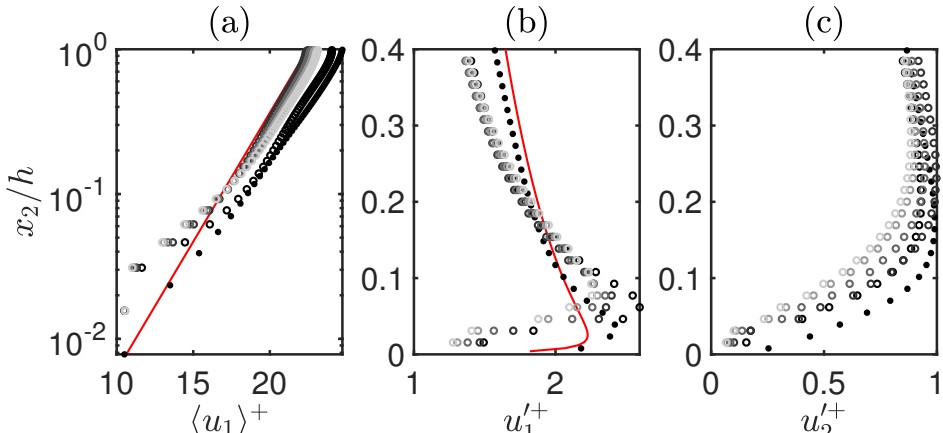

**Figure A1.** Vertical structure of streamwise velocity (a), streamwise velocity RMS (b), vertical velocity RMS (c). Circles, dark to light gray, $C_S = 0.1$ to $C_S = 0.1678$; full circles, PSFD.

## Appendix A

### A1 Smagorinsky constant

A sensitivity analysis on the Smagorinsky constant $C_S$ is here performed, considering $C_S = 0.12$, $C_S = 0.14$, $C_S = 0.16$, and $C_S = 0.1678$ (the default value in OpenFOAM®). All the tests are run on $64^3$ control volumes.

As shown in Fig. A1(a), $C_S$ has an impact on the LLM, whereby the $C_S = 0.1$ case results in the largest positive LLM, in agreement with predictions from the PSFD solver, and larger values of the coefficient predict a smaller, albeit still positive, LLM. The Smagorinsky coefficient has a discernible impact on the velocity RMSs. Specifically, the magnitude of the near-surface maximum in both $u_1'$ (Fig. A1(b)) and $u_2'$ (Fig. A1(c)) is reduced, and its location shifted away from the surface– likely the result of a higher near-surface energy dissipation as $C_S$ is increased. Larger $C_S$ values also yield a more apparent departure from corresponding profiles from the PSFD-based solver.

One-dimensional spectra (Fig. A2(a)) show that larger $C_S$ coefficients result in a more rapid decay of energy density throughout the spectrum, and in a shift of profiles in the inertial sub-range. Interestingly, such profiles are characterized by the same power-law exponent. No value of the Smagorinsky coefficient seems able to yield a $k^{-5/3}$ power law in the inertial sub-range. Further, as also shown in Fig. A2(b) and (c), increasing $C_S$ when compared to the considered value leads to a modest improvement on the $R_{11}$ profiles, with no impact on the previously drawn conclusions.

### A2 Interpolation schemes

The results in §3 made use of the linear interpolation scheme to evaluate the terms in the filtered Navier-Stokes equations at the face-centers, as a consequence of the Gauss divergence theorem. Additional tests were carried out using the QUICK



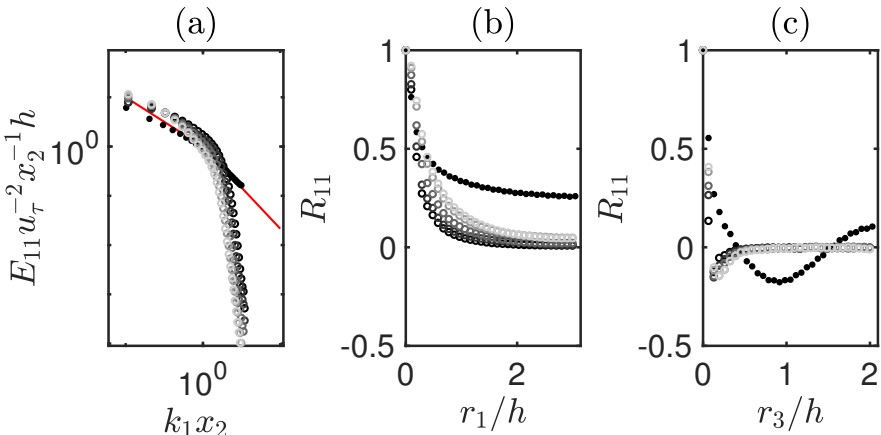

**Figure A2.** Normalized one-dimensional spectrum of streamwise velocity at height $x_2/h \approx 0.1$ (a); one-dimensional spatial autocorrelation of streamwise velocity at height $x_2/h \approx 0.1$ along the streamwise direction (b) and along the spanwise direction (c). Circles, dark to light gray, $C_S = 0.1$ to $C_S = 0.1678$; full circles, PSFD; solid red line, $(k_1 x_2)^{-1}$ in the production range and $(k_1 x_2)^{-5/3}$ in the inertial sub-range.

interpolation scheme (Ferziger and Peric, 2002) for the evaluation of non-linear terms, and results thereof are here compared with the R-$2\pi$ and B-$2\pi$ cases, at the same grid resolution.

Figure A3(a) shows the vertical structure of the mean streamwise velocity. The QUICK and the linear schemes provide the same results in the near-wall region, where an underprediction is observed (see LLM). The interpolation scheme plays a role in the outer layer, where the velocity profile obtained with the QUICK scheme shows a speed-up when compared to the R-$2\pi$ and B-$2\pi$ cases. The RMSs of streamwise and vertical velocities are shown in Fig. A3(b) and (c), respectively. In the near-wall region, an overprediction of $u'_1$ and an underprediction of $u'_2$ characterize the FV results, more severe when the QUICK scheme

is used.

The one-dimensional spectrum, shown in Fig. A4(a), exhibits the $k^{-1}$ power-law behavior typical of the production range at low wavenumber. In the inertial sub-range, the profile obtained with the QUICK scheme decays faster than the one from the B-$2\pi$ case, and the decay starts at lower wavenumbers. In terms of one-dimensional spatial autocorrelation (Fig. A4(b) and (c)), the QUICK interpolation scheme performs slightly better than the linear one, in the sense that the decay of the autocorrelation

is slower.

The instantaneous snapshot of the streamwise velocity fluctuations proposed in Fig. A5, obtained with the QUICK scheme, highlights that the flow field features larger (more coherent) patterns when compared to those shown in Fig. 7.

## A3  `rk4projectionFoam`

In this Sub-Appendix, an alternative solver in the OpenFOAM® framework is considered, and the results are contrasted against

those obtained with `pisoFoam`. The solver, `rk4projectionFoam`, is based on a projection method coupled with the Runge-Kutta 4 time-advancement scheme (Ferziger and Peric, 2002). Details on the implementation can be found in Vuorinen



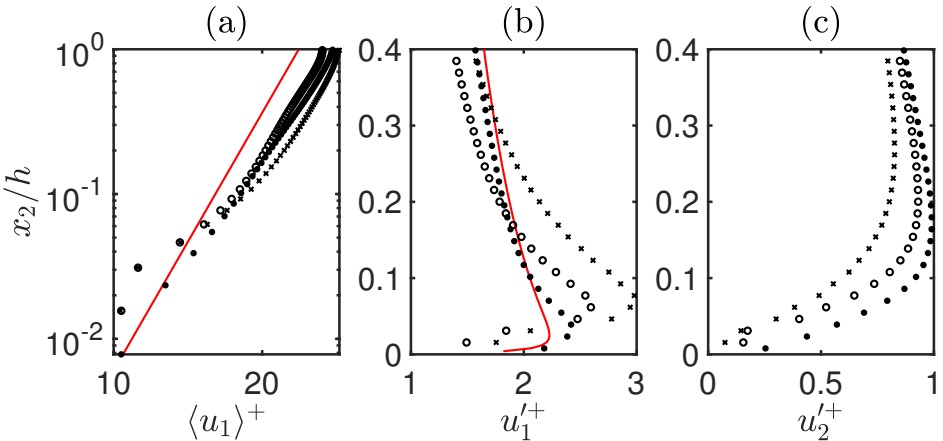

**Figure A3.** Vertical structure of streamwise velocity (a), streamwise velocity RMS (b), vertical velocity RMS (c). Circles, linear interpolation scheme; x-marks, QUICK interpolation scheme; full circles, PSFD.

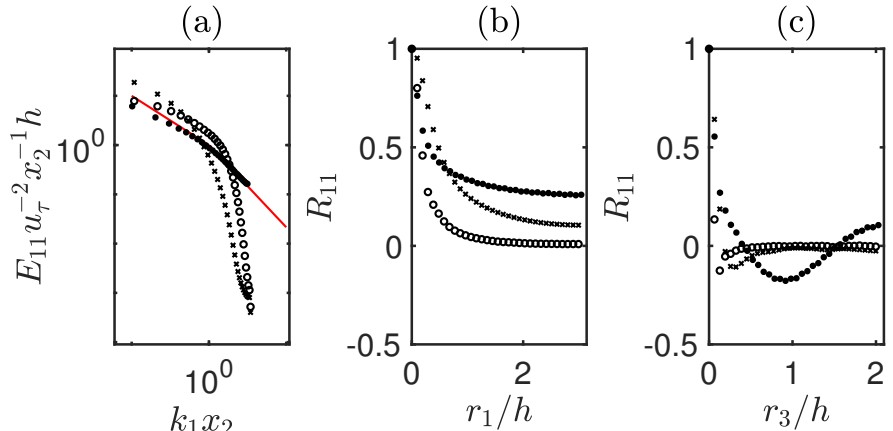

**Figure A4.** Normalized one-dimensional spectrum of streamwise velocity at height $x_2/h \approx 0.1$ (a); one-dimensional spatial autocorrelation of streamwise velocity at height $x_2/h \approx 0.1$ along the streamwise direction (b) and along the spanwise direction (c). Circles, linear interpolation scheme; x-marks, QUICK interpolation scheme; full circles, PSFD.

et al. (2015) (note that in their reported code, a term in the form of a time-step $\Delta t$ is missing, leading to a dimensional mismatch and raising a compile-time error). A comparison of the performances of the solvers has been performed at moderate Reynolds number in Vuorinen et al. (2014), where it is pointed out that `rk4projectionFoam` provides similar results at lower computational cost when compared to `pisoFoam`. In the following, the performances of the solver are tested at high Reynolds number ($Re_\tau = 10^7$). The same cases simulated with `pisoFoam` (Table 1) are considered.

In Fig. A6(a) the vertical profile of the mean streamwise velocity is shown. The `rk4projectionFoam` solver leads to a behavior that is similar to the `pisoFoam` one in the very near surface region, but the profiles feature no LLM in the surface

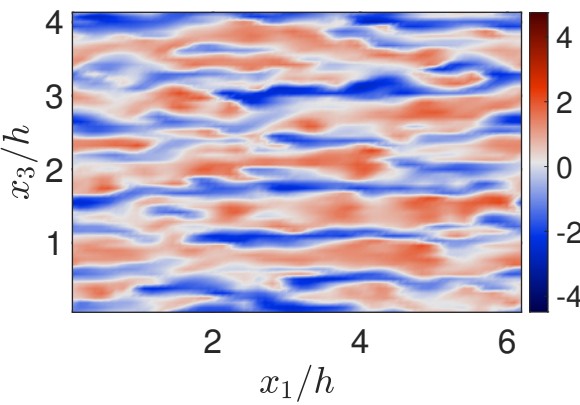

**Figure A5.** Instantaneous snapshot of streamwise velocity fluctuations, as defined in Fig. 7, at height $x_2/h \approx 0.1$.

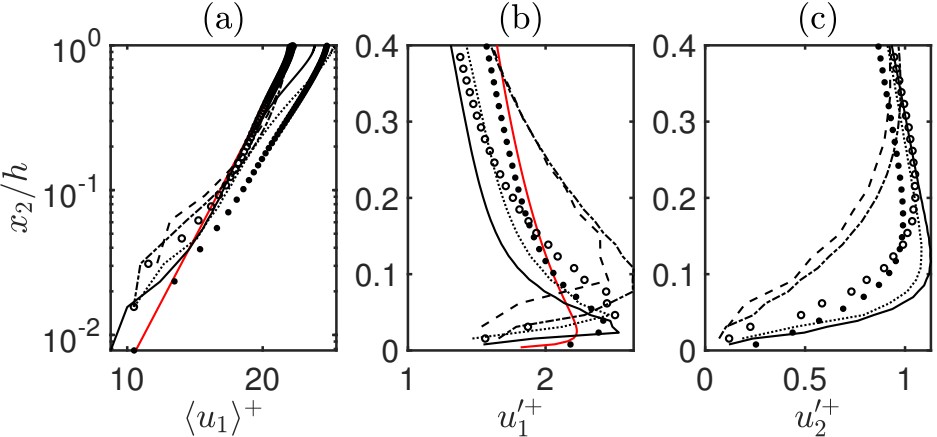

**Figure A6.** Vertical structure of streamwise velocity (a), streamwise velocity RMS (b), vertical velocity RMS (c). Lines and symbols are defined in Table 1.

layer. Streamwise and vertical velocity RMSs are shown in Fig. A6(b) and (c), respectively. The same scenario as the one
obtained with `pisoFoam` is observed: turbulence intensities are underpredicted in the near-wall region, $u_1'^+$-peak values are
overpredicted and $u_2'^+$-peak values are underpredicted (except for the cases B-$\pi$ and F-$2\pi$).

*Author contributions.* BG and MG designed the study, BG conducted the analysis under the supervision of MG, BG and MG wrote the
manuscript.



*Competing interests.* The authors declare that they have no conflict of interest.

*Acknowledgements.* The work was supported via start-up funds provided by the Department of Civil Engineering and Engineering Mechanics at Columbia University. The authors acknowledge computing resources from Columbia University's Shared Research Computing Facility project, which is supported by NIH Research Facility Improvement Grant 1G20RR030893-01, and associated funds from the New York State Empire State Development, Division of Science Technology and Innovation (NYSTAR) Contract C090171, both awarded April 15, 2010. The authors are grateful to Weiyi Li for generating the PSFD data, and to Drs. Ville Vuorinen and George I. Park for useful discussions on
the performance of Finite-Volume-based solvers for the simulation of turbulent flows.



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
