# Peer review of "On the suitability of second-order accurate finite-volume solvers for the simulation of atmospheric boundary layer flow"

_Geoscientific Model Development, 2020_

## Referee Comment (RC1) · Anonymous Referee #1 · 13 Jun 2020

**Review of manuscript GMD-2020-84**

**On the suitability of general-purpose Finite-Volume solvers for the simulation of Atmospheric-Boundary-Layer flow**

Beatrice Giacomini[1] and Marco G. Giometto[1]

[1]Department of Civil Engineering and Engineering Mechanics, Columbia University in the City of New York, 500 W 120th St, New York, NY 10027, USA.

**Recommendation:** revisions

The study described in the reviewed paper definitely makes sense, as it provides valuable insights regarding capabilities of a class of numerical schemes broadly used for simulations of atmospheric boundary-layer flows. It should be specifically mentioned (so far, it is not clear from the title) that techniques specifics are considered only for dynamic part of the problem (that includes Navier-Stokes and continuity equations for incompressible fluid). The study is reported in sufficient detail, and the results are analyzed quite comprehensively and candidly.

The only major issue I have with the study is associated with desperation that the reader feels when – guided by the authors – she/he goes through the figures showing the results and their interpretations, and comes (together with the authors) to a conclusion that the whole situation with application of considered second-order-accurate FV schemes for LES of ABL flows in even basic setup is rather bleak (at resolutions investigated), which brings in question the entire feasibility of such schemes. The authors make some comments on what one may expect from the employed scheme with respect to its ability to reproduce particular features of the ABL turbulent flow. In my view, to make sense out of the paper findings, this discussion needs to be significantly expanded in order to provide the reader with a clear guidance regarding performance of the scheme and explain how its specific deficiencies are associated with its properties.

I have also several minor, mostly editorial, suggestions.

1. Page 1: It should be directly indicated in the title, or at least, in the Abstract and Introduction, that only dynamic subset of ABL governing equations is considered, so that the reader will not have hopes for seeing applications of these solvers for heat and scalar transfer equations.

2. Line 96: "proposed" is a wrong word here.

3. Line 105: Such constancy of density is usually associated with the Boussinesq approximation, which should probably be mentioned directly.

4. Line 122: replace "observation" with "assumption".

5. Line 138: "approximation". This is more correctly called the Boussinesq hypothesis or analogy (to distinguish from the Boussinesq approximation that refers to the density constancy).

6. Line 199: replace "herein proposed" to "presented herein".

7. Line 288: replace "statements" by "findings".

8. Line 333: you need to specify correspondence between the u,v,w notation for velocity components and your standard $u\_1, u\_2, u\_3$ notation (also in other places, where needed).

9. Figure 8: reduce font size of tick labels in $x\_2$ direction.

10. Line 360: speaking of "solvers"; actually, it was a single solver that was investigated.

11. Line 383: the verdict regarding FV-solvers; sounds too general... maybe still not all of them (a grain of optimism)?

---

## Referee Comment (RC2) · Anonymous Referee #2 · 13 Jun 2020

**Review on article with title "On the suitability of general-purpose finite-volume-based solvers for the simulation of atmospheric-boundary-layer flow"**

The paper is a thorough characterization of OpenFOAM for large-eddy simulations of half-channel flows with reference grid sizes $64^3$. In particular, the authors consider a Smagorinsky subgrid-scale model with wall modeling. OpenFOAM is based on a co-located finite volume formulation and the authors compare the results with those obtained from a pseudo-spectral–finite-difference method. The authors study the sensitivity to grid size and grid aspect ratio.

OpenFOAM is an open-source, versatile software tool used for engineering and environmental applications both in industry and academia, including wind-energy applications. The characterization of OpenFOAM when applied to simulate the ABL might therefore be of interest. Nonetheless, I found that some aspects of the manuscript should be strengthen or clarified before publication:

*Major points*

1. This paper is about OpenFOAM rather than a generic finite-volume model. This is stated by the authors in line 195 *"The results herein proposed are hence representative of the OpenFOAM solver with the standard wall-layer treatment – the set-up that is most commonly adopted when using this code"*.

   This information first comes in line 87. The relevance of the paper would be clarified and improved if this information is clearly stated earlier in the manuscript, in the abstract and maybe the title. The abstract should also indicate that the analysis is restricted to the Smagorinsky subgrid-scale model because this is important to interpret the results.

2. The paper considers an "open-channel-flow set-up", and not an atmospheric boundary layer, and this information only comes in line 150. This information should also be in the abstract and the introduction.

3. The sensitivities that the authors study greatly depend on the resolution, measured in terms of the number of grid points (or an effective Reynolds number [Sullivan and Patton, 2011], or the ratio of the filter size to the integral length scale). Therefore, the grid size that the authors consider in the analysis should also be in the abstract, the introduction, and the conclusions. This information is important to interpret the results.

   The sensitivity to grid size is particularly large for the resolutions that the authors consider, which are lower than in common ABL studies. In line 165, the authors write *"The chosen grid resolutions are in line with those typically used in studies of ABL flows (see, e.g., Salesky et al., 2017)."*, but Salesky et al. 2017 uses $160^3$ or $256^3$, which is a substantial difference to $64^3$. Resolution studies consider even larger grid sizes [Sullivan and Patton, 2011].

4. The statements regarding the dependence of the results on resolution are too general. For instance, the authors write

- in line 5, *"It is found that first- and second-order velocity statistics are sensitive to the grid resolution and to the details of the near-wall numerical treatment, and a general improvement is observed with horizontal grid refinement. Higher-order statistics, spectra and autocorrelations of the streamwise velocity, on the contrary, are consistently mispredicted, regardless of the grid resolution."*

- in line 20, *"Although mean flow and second-order statistics become acceptable provided sufficient grid resolution, the use of said solvers might prove problematic for studies requiring accurate higher-order statistics, velocity spectra and turbulence topology."*

- in line 70, *"the excessive damping of resolved-scale energy at high wavenumber is likely to compromise their predictive capabilities for high-Reynolds ABL-flow applications."*

- in line 222, *"Grid refinement in the horizontal directions leads to an improved matching between the FV and the PSFD solver, both in terms of shape and magnitude."*

- in line 233, *"Grid refinement in the horizontal directions improves the matching between the FV-based and the PSFD-based [...]"*

It might be more useful to say how much this dependence on resolution is, i.e., how much one particular property change when changing resolution around a particular value. In the end, as the grid is refined, we would reproduce better and better more and more properties. The important question is what grid size (or effective Reynolds number, or ratio between the filter size and the integral length scale) we need to obtain certain statistics with a given accuracy, in this case, when using OpenFOAM with a Smagorinsky subgrid-scale model in wall-bounded shear flows.

For instance, for direct numerical simulations, we know that second-order methods typically need twice the resolution of spectral methods to similarly represent the variances [Moin and Mahesh, 1998]. What would be the equivalent for OpenFOAM in the model configuration considered in this study?

This comment relates to what the authors write in line 83: *"Note that the studies conducted with FV-based solvers are mainly focused on first- and second-order flow statistics, which are themselves not sufficient to fully characterize turbulence– and related transport– in the ABL."*. What do the authors mean by "fully characterize"? For some applications, correctly representing the first- and second-order moments might be sufficient, whereas for other applications (atmospheric chemistry, wild fires) representing the spectra and LSMs might be insufficient.

5. The introduction reads too much as a review, the focus on OpenFOAM appearing first and unexpectedly in lines 85-90. It might be useful to focus more the introduction around OpenFOAM, the half-channel configuration, and the kind of grid sizes that are considered in this analysis. This might help setting the right expectations earlier in the paper.

In a similar line, the review on LSM between lines 260 to 275 might be shortened.

6. In line 187, the authors indicate that the log-layer mismatch observed in this study is a well-known problem of wall models.

In line 218, the authors indicate that rms-deviations observed in this study is a well-known problem in FV-based WMLES.

What is then new in this manuscript? The particularization to OpenFOAM at this particular resolution? I guess this comment relates to point 1.

*Minor points*

1. In line 137, I am not sure I understand where $u_\tau = \sqrt{\tau_{\alpha 2,w} |\mathbf{u}| / u_\alpha}$ comes from.

   do not understand equations 5 to 6. Related to it "no-slip applies at the lower surface" in line 153 is strange...

2. In line 154, the authors write *"The kinematic viscosity is set to $10^{-7}$ $m^2/s$ in the bulk of the flow, resulting in $Re_\tau = 10^7$"*. I think the information about $Re_\tau$ is meaningless because the effective Reynolds number introduced by the subgrid-scale diffusivity is much smaller. As the authors later say, one can neglect the molecular viscosity against the subgrid-scale viscosity. The value of the viscosity is also a bit strange for an ABL context.

3. Adding colors in the figures might help the reader to distinguish the various cases more easily.

4. In line 227, the authors refer to the results of del Alamo et al 2006 regarding skewness, flatness and correlation coefficient. It migtht be useful to add this data to figure 3.

5. In table 3, why taking the tangent point to $\kappa^{-5/3}$ to distinguish between inertial and large-scale and not some integral length scale [Pope, 2000]? Moreover, 32 points seem too few to distinguish an inertial subrange.

**References**

P. Moin and K. Mahesh. Direct numerical simulation: A tool in turbulence research. *Annu. Rev. Fluid Mech.*, 30:539–578, 1998.

S. B. Pope. *Turbulent Flows.* Cambridge University Press, 2000.

P. P. Sullivan and E. G. Patton. The effect of mesh resolution on convective boundary layer statistics and structures generated by large-eddy simulations. *J. Atmos. Sci.*, 68:2395–2415, 2011.

---

## Author Comment (AC1) · 2 Oct 2020

**1 Response to reviewer 1**

We thank the reviewer for his/her time and for the constructive comments, which helped improve the quality of the manuscript. We address each comment below.

**1.1 Major comments**
* * *
***Reviewer statement 1***: *The study described in the reviewed paper definitely makes sense, as it provides valuable insights regarding capabilities of a class of numerical schemes broadly used for simulations of atmospheric boundary-layer flows. It should be specifically mentioned (so far, it is not clear from the title) that techniques specifics are considered only for dynamic part of the problem (that includes Navier–Stokes and continuity equations for incompressible fluid). The study is reported in sufficient detail, and the results are analyzed quite comprehensively and candidly.*

**Response**: We thank the reviewer for this comment. In the Abstract it is now explicitly stated that a neutrally-stratified ABL flow without Coriolis effects is considered.
* * *
***Reviewer statement 2***: *The only major issue I have with the study is associated with desperation that the reader feels when—guided by the authors—she/he goes through the figures showing the results and their interpretations, and comes (together with the authors) to a conclusion that the whole situation with application of considered second-order-accurate FV schemes for LES of ABL flows in even basic setup is rather bleak (at resolutions investigated), which brings in question the entire feasibility of such schemes. The authors make some comments on what one may expect from the employed scheme with respect to its ability to reproduce particular features of the ABL turbulent flow. In my view, to make sense out of the paper findings, this discussion needs to be significantly expanded in order to provide the reader with a clear guidance regarding performance of the scheme and explain how its specific deficiencies are associated with its properties.*

**Response**: We thank the reviewer for this critical input. If the reviewer refers to a detailed, applied-math type analysis focusing on the impact of discretization and modeling errors on the solution, one would then need to introduce strong simplifications to make such an analysis possible (see e.g. [1] and [2]) and findings might not then be directly transferrable to the ABL flow system. The aim of this work is a different one, namely to analyze the performance of a class of general-purpose FV solvers in the *full-fledged* setup for the study of ABL flows, with a focus on their capabilities to predict physical quantities that are of interest for the ABL community without necessarily ascribing limitations thereof to specific properties of the numerical scheme. This type of

analysis is within the aim and scope of this journal, which lists "[...]full evaluations of previously published models" as one of the manuscripts' categories (see `https://www.geoscientific-model-development.net/about/manuscript_types.html`). In an efford to address the comment from the reviewer while keeping the analysis limited to the physics of the system, we identified an important limitation of the considered class of FV-solvers, which sheds light on some of the observed discrepancies and variability in flow statistics. Specifically, this class of FV solvers is not able to correctly capture the dominant momentum transport mechanism in the ABL, namely the sweep and ejection pairs, at the considered resolutions (see discussion in §3.3 as well as related comments in the Abstract and Conclusions sections of the revised manuscript, which we also report below).

> "(Abstract) At the considered resolutions, the considered class of FV-based solvers yields a poorly correlated flow field and is not able to accurately capture the dominant mechanisms responsible for momentum transport in the ABL, especially when using linear interpolation schemes for the discretization of non-linear terms. The latter consist of sweeps and ejection pairs organized side by side along the cross-stream direction, representative of a streamwise roll mode. This shortcoming leads to a misprediction of flow statistics that are relevant for ABL applications and to an enhanced sensitivity of the solution to variations in grid resolution, calling for future research aimed at reducing the impact of modeling and discretization errors."

> "(Section 3.3) As apparent from Fig. 9, the PSFD conditionally-averaged velocity field exhibits counter-rotating patterns associated with positive and negative streamwise velocity fluctuations (corresponding to the aforementioned streaks). The roll modes feature a diameters ($d \approx h$) throughout the ABL, which is consistent with findings from the literature, and positive an negative velocity fluctuations are approximately of the same magnitude ($\approx u_\tau$). From Fig. 10, it is also apparent how the considered isosurfaces extend in the streamwise direction for about $4h$. Quite surprisingly, the FV-based solver is not able to predict the roll modes, irrespective of the interpolation scheme or resolution, and the magnitude of the low-momentum streaks is also severely underpredicted across the considered cases. Further, Figs. 1 and 8 both depict a FV conditionally-averaged flow field that is poorly correlated in the cross-stream and streamwise directions, resulting in significantly smaller momentum-carrying structures. This supports previous findings from the two-point correlation maps (Fig. 4). The lack of roll modes implies that these solvers are not able to capture the fundamental mechanism supporting momentum transfer in the ABL, at least at the considered grid resolutions. This limitation can also be identified as the root cause of several of the observed problematics with the FV solutions, including the relatively high (low) streamwise-velocity skewness when using using linear (QUICK) schemes (see Fig. 2,a) and the inbalance between sweeps and ejections (Fig. 1

and Fig. 8)."

"(Conclusions) To summarize, the considered class of FV-based solvers overall predicts a flow field that is less correlated in space when compared that of the PSFD solver and is not able to capture the salient features responsible for momentum transfer in the ABL, at least at the considered grid resolutions. These limitations appear to be the root cause of many of the observed discrepancies between FV flow statistics and the reference PSFD or experimental ones, including the mispredicted streamwise-velocity skewness (Fig. 2,a), the inbalance between sweeps and ejections (Fig. 1 and Fig. 8), and the overall sensitivity of flow statistics to variations in the grid resolution."

**1.2 Minor comments**
* * *
***Reviewer statement 1***: *Page 1: It should be directly indicated in the title, or at least, in the Abstract and Introduction, that only dynamic subset of ABL governing equations is considered, so that the reader will not have hopes for seeing applications of these solvers for heat and scalar transfer equations.*

**Response**: Please see the response to the statement 1 in the Major comments section.
* * *
***Reviewer statement 2***: *Line 96: "proposed" is a wrong word here.*

**Response**: The sentence was edited as "Results are shown in §3 ..."
* * *
***Reviewer statement 3***: *Line 105: Such constancy of density is usually associated with the Boussinesq approximation, which should probably be mentioned directly.*

**Response**: This comment was addressed as follows: "... $\rho$ is the (constant, under the Boussinesq approximation) fluid density, ..."
* * *
***Reviewer statement 4***: *Line 122: replace "observation" with "assumption".*

**Response**: The sentence now reads: "This conjecture is supported by the results of Majander and Siikonen (2002)."
* * *
***Reviewer statement 5***: *Line 138: "approximation". This is more correctly called the Boussinesq hypothesis or analogy (to distinguish from the Boussinesq approximation that refers to the density constancy).*

**Response**: "... (Boussinesq approximation)..." was replaced by "... (Boussinesq hypothesis)..."
* * *
***Reviewer statement 6***: *Line 199: replace "herein proposed" to "presented herein".*

**Response**: The paragraph has been rearranged and the sentence that was at line 199 does not appear anymore.
* * *
***Reviewer statement 7***: *Line 288: replace "statements" by "findings".*

**Response**: This comment was addressed and the sentence reads: "The one-dimensional spatial autocorrelation ($R_{uu}$), shown in Fig. 5 along the streamwise and cross-stream directions, further corroborates the above findings"
* * *
***Reviewer statement 8***: *Line 333: you need to specify correspondence between the $u, v, w$ notation for velocity components and your standard $u_1, u_2, u_3$ notation (also in other places, where needed).*

**Response**: Instead of specifying a correspondence between $u, v, w$ and $u_1, u_2, u_3$, the notation was unified consistently with the rest of the paper (where the subscripts $x, y, z$ are used to denote streamwise, cross-stream, vertical directions, respectively, and correspondent vectorial components, and $(u, v, w) = (u_x, u_y, u_z)$).
* * *
***Reviewer statement 9***: *Figure 8: reduce font size of tick labels in $x_2$ direction.*

**Response**: The font size was reduced.
* * *
***Reviewer statement 10***: *Line 360: speaking of "solvers"; actually, it was a single solver that was investigated.*

**Response**: The "numerical framework" is indeed a single one, namely OpenFOAM®. Within this single framework, different *numerical procedures* have been considered, varying the pressure-velocity coupling method (PISO vs fractional step method), the time stepping scheme (Adam-Moulton vs Runge Kutta), and the linear interpolation scheme for the non-linear terms (linear vs QUICK). We regarded each of these procedures as "solvers", which is why we referred to "solvers". This is also standard terminology in the OpenFOAM community. Note that in the revised version of the manuscript the main analysis has now been extended to additional "solvers", and is not anymore predominantly focused on the PISO algorithm with linear interpolation scheme. We have also relaxed the terminology and we now refer to "[. . .]an important class of general-purpose, second order accurate FV solvers."
* * *
***Reviewer statement 11***: *Line 383: the verdict regarding FV-solvers; sounds too general. . . maybe still not all of them (a grain of optimism)?*

**Response**: We thank the reviewer for this comment and totally agree with it. We have tested only one specific class of FV solvers, and alternative ones such as those based on staggered grid setups or higher order discretization schemes have been shown to feature improved conservation properties and behaviors for high Reynolds number flows. We now made an effort to point out that findings proposed herein only pertain to the specific class of FV solvers that was considered (see below). We also mentioned that approaches based on a staggered grid setup might be required to improve the quality of predictions.

Abstract: "The present work assesses the quality and reliability of an important class of general-purpose, second-order accurate finite-volume (FV) solvers in the large-eddy simulation of a neutrally-stratified atmospheric boundary layer (ABL) flow."

Conclusions: "This work provides insight on the quality and reliability of an important class of general-purpose, second-order accurate FV-based solvers in wall-modeled LES of neutrally-stratified ABL flow. The FV solvers are part of the OpenFOAM® framework, make use of the divergence form of the nonlinear term, and are based on a colocated arrangement for the evaluation of the unknowns."

Conclusions: "To summarize, the considered class of FV-based solvers overall predicts a flow field that is less correlated in space when compared that of the PSFD solver and is not able to capture the salient features responsible for momentum transfer in the ABL, at least at the considered grid resolutions."

**References**

[1] S. Ghosal. An Analysis of Numerical Errors in Large-Eddy Simulations of Turbulence. *Journal of Computational Physics*, 125(1):187–206, apr 1996.

[2] J. Meyers, P. Sagaut, and B. J. Geurts. Optimal model parameters for multi-objective large-eddy simulations. *Physics of Fluids*, (9), 2006.

---

## Author Comment (AC2) · 2 Oct 2020

**1 Response to Reviewer 2**

We thank the reviewer for his/her time and for the constructive comments, which helped improve the quality of the manuscript. We address each comment below.

**1.1 Major comments**
* * *
**Reviewer statement 1a**: *This paper is about OpenFOAM rather than a generic finite-volume model. This is stated by the authors in line 195 "The results herein proposed are hence representative of the OpenFOAM solver with the standard wall-layer treatment—the set-up that is most commonly adopted when using this code". This information first comes in line 87. The relevance of the paper would be clarified and improved if this information is clearly stated earlier in the manuscript, in the abstract and maybe the title.*

**Response**: We thank the reviewer for this comment. We have now pointed out in the abstract, introduction and conclusions that the "class of solvers" considered herein are based on the OpenFOAM "framework" (see quotations below). We refrained from adding the *OpenFOAM* keyword in the title because findings from this study are not limited to solvers built within OpenFOAM but extend to any finite volume software/code/solver relying on these same discretization and physical-modeling procedures.

> "(Abstract) The present work assesses the quality and reliability of an important class of general-purpose, second-order accurate finite-volume (FV) solvers in the large-eddy simulation of a neutrally-stratified atmospheric boundary layer (ABL) flow. [...] Simulations are carried out within the OpenFOAM® framework, which is based on a colocated grid arrangement."

> "(Introduction) Motivated by the aforementioned needs, the present study aims at characterizing the quality and reliability of an important class of second-order accurate FV solvers for the LES of neutrally-stratified ABL flows. The analysis is conducted in the open-channel flow setup (no Coriolis acceleration) via the OpenFOAM® framework (Weller et al., 1998; De Villiers, 2006; Jasak et al., 2007)."

> "(Conclusions) [...]This work provides insight on the quality and reliability of an important class of general-purpose, second-order accurate FV-based solvers in wall-modeled LES of neutrally-stratified ABL flow. The FV solvers are part of the OpenFOAM® framework, make use of the divergence form of the nonlinear term, and are based on a colocated arrangement for the evaluation of the unknowns."

We've also add a dedicated discussion in the Methodology section on the characteristics of the OpenFOAM framework, that are shared by the considered class of solvers:

> "(Methodology) In the OpenFOAM® framework, the computational grid is colocated. Although advantageous in complex domains when compared to staggered grids (Ferziger and Peric, 2002), the colocated arrangement is known to cause difficulties with pressure-velocity coupling, hence requiring specific procedures to avoid oscillations in the solution. The standard Rhie-Chow correction (Rhie and Chow, 1983) is here adopted, which is known to negatively affeect the energy-conservation properties of central schemes (Ferziger and Peric, 2002). In addition, when approximating the integrals over the surfaces bounding each control volume (as a consequence of the Gauss divergence theorem), the unknowns are evaluated at face-centers and are assumed to be constant at each face, yielding an overall second-order spatial accuracy (Churchfield et al., 2010). Since the divergence form of the convective term is used in combination with a low-order scheme over a non-staggered grid, the solution is inherently unstable (Kravchenko and Moin, 1997). The present work makes use of the linear and the QUICK interpolation schemes (Ferziger and Peric, 2002) to evaluate the unknowns at face-centers. The numerical solver combines the PISO algorithm (Issa, 1985) for the pressure-velocity calculation and an implicit Adams–Moulton scheme for time integration (Ferziger and Peric, 2002). The performances of an alternative solver characterized by a Runge–Kutta time-advancement scheme and a projection method for the pressure-velocity coupling (Vuorinen et al., 2014) are also analyzed in the Appendix."
* * *
*Reviewer statement 1b*: *The abstract should also indicate that the analysis is restricted to the Smagorinsky subgrid-scale model because this is important to interpret the results.*

**Response**: This comment was addressed by adding the following sentence to the Abstract:

> "A series of open-channel flow simulations are performed using a static Smagorinsky model for sub-grid scale momentum fluxes and an algebraic equilibrium wall-layer model."
* * *
*Reviewer statement 2*: *The paper considers an "open-channel-flow set-up", and not an atmospheric boundary layer, and this information only comes in line 150. This information should also be in the abstract and the introduction.*

**Response**: This point was addressed by adding the following to the Abstract:

> "The analysis is carried out within the OpenFOAM® framework, which is based on a colocated grid arrangement. The spatially-filtered incompressible Navier–Stokes equations are solved in an open-channel flow setup (no Coriolis acceleration)."

and the following to the Introduction:

> "The analysis is conducted in the open-channel flow setup (no Coriolis acceleration) via the OpenFOAM® framework (Weller et al., 1998; De Villiers, 2006; Jasak et al., 2007)."
* * *
***Reviewer statement 3***: *The sensitivities that the authors study greatly depend on the resolution, measured in terms of the number of grid points (or an effective Reynolds number [Sullivan and Patton, 2011], or the ratio of the filter size to the integral length scale). Therefore, the grid size that the authors consider in the analysis should also be in the abstract, the introduction, and the conclusions. This information is important to interpret the results. The sensitivity to grid size is particularly large for the resolutions that the authors consider, which are lower than in common ABL studies. In line 165, the authors write "The chosen grid resolutions are in line with those typically used in studies of ABL flows (see, e.g., Salesky et al., 2017).", but Salesky et al. 2017 uses $160^3$ or $256^3$, which is a substantial difference to $64^3$. Resolution studies consider even larger grid sizes [Sullivan and Patton, 2011].*

**Response**: We agree with the point made by the reviewer. This comment was addressed by extending the analysis to cases up to $160^3$ control volumes. The manuscript was edited accordingly throughout. We want to point out that while $160^3$ collocation nodes in Cartesian codes run in a matter of hours for ABL flow simulations, FV-solvers supporting unstructured grid approaches are almost two orders of magnitude slower and $160^3$ hence represents a state-of-the-art resolution. Overall though we agree that this resolution is relatively modest and hence results are expected to depend on this factor. This aspect of the problem has been pointed out in the Results section of the revised manuscript and in the Conclusions.
* * *
***Reviewer statement 4***: *The statements regarding the dependence of the results on resolution are too general. For instance, the authors write*

> *in line 5, "It is found that first- and second-order velocity statistics are sensitive to the grid resolution and to the details of the near-wall numerical treatment, and a general improvement is observed with horizontal grid refinement. Higher-order statistics, spectra and autocorrelations of the streamwise*

*velocity, on the contrary, are consistently mispredicted, regardless of the grid resolution."*

*in line 20, "Although mean flow and second-order statistics become acceptable provided sufficient grid resolution, the use of said solvers might prove problematic for studies requiring accurate higher-order statistics, velocity spectra and turbulence topology."*

*in line 70, "the excessive damping of resolved-scale energy at high wavenumber is likely to compromise their predictive capabilities for high-Reynolds ABL-flow applications."*

*in line 222, "Grid refinement in the horizontal directions leads to an improved match- ing between the FV and the PSFD solver, both in terms of shape and magnitude."*

*in line 233, "Grid refinement in the horizontal directions improves the matching between the FV-based and the PSFD-based [. . . ]"*

*It might be more useful to say how much this dependence on resolution is, i.e., how much one particular property change when changing resolution around a particular value. In the end, as the grid is refined, we would reproduce better and better more and more properties. The important question is what grid size (or effective Reynolds number, or ratio between the filter size and the integral length scale) we need to obtain certain statistics with a given accuracy, in this case, when using OpenFOAM with a Smagorinsky subgrid-scale model in wall-bounded shear flows. For instance, for direct numerical simulations, we know that second-order methods typically need twice the resolution of spectral methods to similarly represent the variances [Moin and Mahesh, 1998]. What would be the equivalent for OpenFOAM in the model configuration considered in this study? This comment relates to what the authors write in line 83: "Note that the studies conducted with FV-based solvers are mainly focused on first- and second-order flow statistics, which are themselves not sufficient to fully characterize turbulence—and related transport—in the ABL.". What do the authors mean by "fully characterize"? For some applications, correctly representing the first- and second-order moments might be sufficient, whereas for other applications (atmospheric chemistry, wild fires) representing the spectra and LSMs might be insufficient.*

**Response**:

We thank the reviewer for this critical input. We devoted a significant amount of efforts to this task. Both relative variations of FV profiles as a function of resolution and variations of the FV profiles with respect to reference PSFD and/or experimental ones at different resolutions were evaluated. We concluded that experimental profiles represent a good candidate for the convergence analysis at these relatively low resolutions where convergence is not strictly related to the order of accuracy of the scheme and is often non monotonic. The Results section now features several tables with quantitative measures of convergence of selected flow statistics against corresponding experimental values (streamwise velocity variances, velocity skewness, kurtosis, and integral length

scales). Note that the convergence is non-monotonic in most cases, due to the modest resolution and to the interaction of discretization and physical modeling errors, whose quantification is not trivial for this complex flow system (Meyers et al., 2006; Meyers and B.J. Geurts, 2007; Meyers and Sagaut, 2007; Ghosal, 1996). These tables nevertheless provide useful insight on the performance of FV-based solver and more quantitative information the community will benefit from. It is also apparent that higher grid resolutions are required for FV solvers to match results from the PSFD solver, or to at the least capture the dominant momentum transfer mechanisms in the channel flow system. The latter was identified as the main limitation of the considered class of FV solvers. Given that general-purpose FV solvers supporting unstructured grid setups are typically two orders of magnitude slower than PSFD solvers, going well beyond $160^3$ control volumes will be a rather challenging task, and justifies the proposed study.

W ehave removed the following comment in the revised version of the manuscript: "Note that the studies conducted with FV-based solvers are mainly focused on first- and second-order flow statistics, which are themselves not sufficient to fully characterize turbulence—and related transport—in the ABL.".
* * *
***Reviewer statement 5****: The introduction reads too much as a review, the focus on OpenFOAM appearing first and unexpectedly in lines 85-90. It might be useful to focus more the introduction around OpenFOAM, the half-channel configuration, and the kind of grid sizes that are considered in this analysis. This might help setting the right expectations earlier in the paper. In a similar line, the review on LSM between lines 260 to 275 might be shortened.*

**Response**: We thank the reviewer for this input. We have streamlined the Introduction, which now provides the motivation and objectives of the work. Details regarding the setup of the problem such as the considered grid resolutions, the half channel configuration, etc. are also briefly mentioned, but a detailed discussion of these aspects is postponed to the Methodology section (see revised manuscript).
* * *
***Reviewer statement 6****: In line 187, the authors indicate that the log-layer mismatch observed in this study is a well-known problem of wall models. In line 218, the authors indicate that rms-deviations observed in this study is a well-known problem in FV-based WMLES. What is then new in this manuscript? The particularization to OpenFOAM at this particular resolution? I guess this comment relates to point 1.*

**Response**: We thank again the reviewer for this critical input. To the author's knowledge, this is the first assessment of the performance of this important class of FV-based solvers for the simulation of ABL flows at this level of detail/insight from a flow

physics perspective. For example, no study has previously assessed the capability of second-order accurate FV-based solvers in capturing momentum transfer mechanisms in ABL flows, how this depends on details of the numerical discretization and how it impacts relevant flow statistics. One of the key novel findings is "[...]that this class of FV-based solvers overall predicts a flow field that is less correlated in space when compared that of the PSFD solver and is not able to capture the salient features responsible for momentum transfer in the ABL, at least at the considered grid resolutions. These limitations appear to be the root cause of many of the observed discrepancies between FV flow statistics and the reference PSFD or experimental ones, including the mispredicted streamwise-velocity skewness, the inbalance between sweeps and ejections, and the overall sensitivity of flow statistics to variations in the grid resolution." With regards to the log-layer mismatch and rms-deviations: Most of the previous findings pertained to relatively low Reynolds numbers. Here we have shown that these problematics are a problem also at ABL Reynolds numbers and that procedures devised to alleviate the log-layer mismatch issue do not seem to work, which motivates further research in the field. We also showed how this mismatch depends on the numerical scheme that is used and how it depends to grid resolution, which is new. Note also that the revised manuscript has been substantially modified and now includes a detailed comparison between the performance of two interpolation schemes for the discretization of nonlinear terms, and how these schemes affect the above quantities has been commented. Further, in an efford to address this comment from the reviewer, the main contributions of this work have now been listed in the Conclusions section of the revised manuscript.

**1.2 Minor comments**
* * *
**Reviewer statement 1**: *In line 137, I am not sure I understand where $u_\tau = \sqrt{\tau_{\alpha 2,w}|\boldsymbol{u}|/u_\alpha}$ comes from. I do not understand equations 5 to 6. Related to it "no-slip applies at the lower surface" in line 153 is strange...*

**Response**: Sub-section 2.1 was expanded to provide a more detailed derivation of the wall-model. Specifically, the following lines were added:

"Employing the no-slip condition for the velocity field, the standard FV approximation of the shear stress at the wall gives (Mukha et al., 2019)

$$\tau_{i2,w} = (\nu + \nu_t)\frac{\partial u_i}{\partial x_2}\Big|_w, \quad i = 1, 3 \ ,$$

where the subscript $f$ is used to denote the evaluation at the center of the wall face, the subscript $c$ denotes the evaluation at the center of the wall adjacent cell and $\Delta x_2$ is the distance from the wall. From the logarithmic law (Eq. 4) evaluated at the first cell-center, one can write $u_\tau = \kappa|\mathbf{u}|_c/\ln(\frac{\Delta x_2}{x_{2,0}})$. Using the definition of friction velocity $u_\tau = \sqrt{\tau_w^2}$, where $\tau_w$ is the magnitude of

the wall shear stress vector, along with Eq. 5, and rearranging, the total
eddy viscosity at the wall reads..."

The sentence "... no-slip applies at the lower surface..." refers to the no-slip condition employed in combination with the wall-model.
* * *
***Reviewer statement 2***: *In line 154, the authors write "The kinematic viscosity is set to $10^{-7}$ m$^2$/s in the bulk of the flow, resulting in $Re = 10^7$". I think the information about Re is meaningless because the effective Reynolds number introduced by the subgrid-scale diffusivity is much smaller. As the authors later say, one can neglect the molecular viscosity against the subgrid-scale viscosity. The value of the viscosity is also a bit strange for an ABL context.*

**Response**: We agree with the comment. However, the simulations were run by setting the kinematic viscosity at $10^{-7}$ m$^2$/s. For this reason, the sentence was edited as follows:

"The kinematic viscosity is set to a nominal value of $10^{-7}$ m$^2$/s, which results in an essentially inviscid flow."
* * *
***Reviewer statement 3***: *Adding colors in the figures might help the reader to distinguish the various cases more easily.*

**Response**: Colors were added to the figures.
* * *
***Reviewer statement 4***: *In line 227, the authors refer to the results of Del Alamo et al. 2006 regarding skewness, flatness and correlation coefficient. It might be useful to add this data to figure 3.*

**Response**: Del Alamo et al. 2006 do not refer to quantitative data. In Fig. 3, the measurements from Monty et al. (2009) were added.
* * *
***Reviewer statement 5***: *In table 3, why taking the tangent point to $k^{-5/3}$ to distinguish between inertial and large-scale and not some integral length scale [Pope, 2000]? Moreover, 32 points seem too few to distinguish an inertial subrange.*

**Response**: We thank the reviewer for this input. We agree that leveraging integral length scales would be a preferrable approach to distinguish between the inertial and the

production range. However, depending on the numerical setup, FV-based solvers at the considered resolutions are severely underpredicting integral length scales (see Tab. 4), thus complicating the analysis / interpretation of results. In view of this limitation, and as part of a paper-streamlining effort, we have removed this PSD analysis along with Table 3.

**References**

Churchfield, M., Vijayakumar, G., Brasseur, J., and Moriarty, P. (2010). Wind energy-related atmospheric boundary layer large-eddy simulation using OpenFOAM. Presented as Paper 1B.6 at the American Meteorological Society, 19[th] Symposium on Boundary Layers and Turbulence NREL/CP-500-48905, National Renewable Energy Laboratory, Colorado.

De Villiers, E. (2006). *The potential for large eddy simulation for the modeling of wall bounded flows*. PhD thesis, Imperial College of Science, Technology and Medicine.

Ferziger, J. and Peric, M. (2002). *Computational methods for fluid dynamics*. Springer.

Ghosal, S. (1996). An analysis of numerical errors in large-eddy simulations of turbulence. *J. Comput. Phys.*, 125:187–206.

Issa, R. (1985). Solution of the implicitly discretised fluid flow equations by operator-splitting. *J. Comput. Phys.*, 62:40–65.

Jasak, H., Jemcov, A., and Tukovic, Z. (2007). OpenFOAM: A c++ library for complex physics simulations. Presented at the International Workshop on Coupled Methods in Numerical Dynamics, IUC, Dubrovnik, Croatia.

Kravchenko, A. and Moin, P. (1997). On the effect of numerical errors in large eddy simulations of turbulent flows. *J. Comput. Phys.*, 131:310–322.

Meyers, J. and B.J. Geurts, P. S. (2007). A computational error-assessment of central finite-volume discretizations in large-eddy simulation using a smagorinsky model. *J. Comput. Phys.*, 227:156–173.

Meyers, J. and Sagaut, P. (2007). Is plane-channel flow a friendly case for the testing of large-eddy simulation subgrid-scale models? *Phys. Fluids*, 19.

Meyers, J., Sagaut, P., and Geurts, B. (2006). Optimal model parameters for multi-objective large-eddy simulations. *Phys. Fluids*, 18.

Rhie, C. and Chow, W. (1983). Numerical study of the turbulent flow past an airfoil with trailing edge separation. *AIAA J.*, 21:1525–1532.

Vuorinen, V., Keskinen, J.-P., Duwig, C., and Boersma, B. (2014). On the implementation of low-dissipative Runge-Kutta projection methods for time dependent flows using OpenFOAM[®]. *Comput. Fluids*, 93:153–163.

Weller, H., Tabor, G., Jasak, H., and Fureby, C. (1998). A tensorial approach to computational continuum mechanics using object-oriented techniques. *Comput. Phys.*, 12:620–631.

---

## Referee Report (RR1)

**On the suitability of second-order accurate finite-volume solvers for the simulation of atmospheric boundary layer flow**

Beatrice Giacomini1 and Marco G. Giometto1

1Department of Civil Engineering and Engineering Mechanics, Columbia University in the City of New York, 500 W 120th St, New York, NY 10027, USA.

**Recommendation:** minor revisions**

The manuscript has been significantly improved compared to the original version. Now it documents even more convincingly that, at investigated grid resolutions, FV-based solvers of the considered class are "not able to accurately capture the dominant mechanisms responsible for momentum transport" in neutrally stratified atmospheric boundary layer flows — a rather discouraging, albeit just, conclusion.

While reading the revised manuscript, I found (I should had noticed this earlier) that governing equations (1), (2), and subsequent material are presented in a very confusing manner, using notation that does not make sense to me.

Assuming that  $\nabla$  is a regular del operator ( $\nabla = \mathbf{i} \frac{\partial}{\partial x} + \mathbf{j} \frac{\partial}{\partial y} + \mathbf{k} \frac{\partial}{\partial z}$ ), Eq. 1 for the nondivergent filtered velocity field,  $\nabla \cdot \mathbf{u} = 0$ , is fine, but what does  $\nabla \mathbf{u} \cdot \mathbf{u}$  in (2) then mean? Operator  $\nabla$  does not apply to a vector  $\mathbf{u}$  as  $\nabla \mathbf{u}$ , so apparently  $\nabla \mathbf{u} \cdot \mathbf{u}$  is supposed to read  $\nabla (\mathbf{u} \cdot \mathbf{u})$ , but this is wrong, as one expects this term to be  $(\mathbf{u} \cdot \nabla)\mathbf{u}$  (provided  $\nabla \cdot \mathbf{u} = 0$ ).

Furthermore,  $\nabla$  is also applied to tensors ( $\tau$  and  $\tau^{SGS,dev}$ ) as  $\nabla \cdot \tau$  and  $\nabla \cdot \tau^{SGS,dev}$ , while the operation  $\nabla \cdot$  for tensors is generally not defined. Besides this, terms  $\nabla \cdot \tau$  and  $\nabla \cdot \tau^{SGS,dev}$  (whatever they mean) must enter (2) with the same sign, as  $\tau$  and  $\tau^{SGS,dev}$  are quantities of the same physical nature (kinematic stresses). But in fact, concluding from how these variables are defined,  $\tau = -2\nu S$  and  $\tau^{SGS,dev} = -2\nu^{SGS} S$ , they are rather kinematic momentum fluxes (negative of stresses, cf. Eqs. 5 and 6).

This has to be straightened out and corrected, if needed. Maybe, presenting Eqs. 1 and 2 in tensor (rather than in vector) notation will help to make things clearer. Hopefully, for simulations the discretized equations have been employed in their correct forms.

Another problem I have is related to the choice of notation for horizontal velocity vector in (4). Notation **u** is already reserved for the 3D filtered velocity vector in Eqs. 1 and 2, which implies

that  $|\mathbf{u}|$  can be nothing else than  $\sqrt{u^2 + v^2 + w^2}$ , so it would be necessary to introduce a horizontal 2D velocity vector  $\mathbf{v} = (u, v)$  with  $|\mathbf{v}| = \sqrt{u^2 + v^2}$  and make corresponding adjustments in the remaining equations of Sect. 2.1.

Other minor points.

- 1. Table 1. Total number of grid/cell points (e.g., 1603) is not a measure of grid resolution.
- 2. Figure 1. Primes (') are typically reserved for denoting fluctuations, not their RMS values. May be a source of confusion.
- 3. Table 2. Are you sure that you need four decimal places to characterize the relative error?
- 4. Table 3. Same problem. Here you even use five decimal places...
- 5. Table 4. See two previous points.

6. Line 294. You need to be more specific about the way double-primed quantities have been evaluated and comment on meaning of their signs.

7. Line 321. Apparently, it should be  $C_S = 0.1678$ .

---

## Author Response (AR2)

**1 Response to reviewer 1**

We thank the reviewer once again for his feedback, which has helped significantly improve the quality of the manuscript.

**1.1 Major comments**
* * *
***Reviewer statement 1***: *Assuming that $\nabla$ is a regular del operator ($\nabla = \mathbf{i}\frac{\partial}{\partial x} + \mathbf{j}\frac{\partial}{\partial y} + \mathbf{k}\frac{\partial}{\partial z}$), Eq. 1 for the nondivergent filtered velocity field, $\nabla \cdot \mathbf{u} = 0$, is fine, but what does $\nabla\mathbf{u} \cdot \mathbf{u}$ in (2) then mean? Operator $\nabla$ does not apply to a vector $\mathbf{u}$ as $\nabla\mathbf{u}$, so apparently $\nabla\mathbf{u} \cdot \mathbf{u}$ is supposed to read $\nabla(\mathbf{u} \cdot \mathbf{u})$, but this is wrong, as one expects this term to be $(\mathbf{u} \cdot \nabla)\mathbf{u}$ (provided $\nabla \cdot \mathbf{u} = 0$).*

**Response**: We thank the reviewer for this comment. Overall it depends on how the $\cdot$ operator is defined, but we agree with that the chosen notation was confusing and we now switched to the more standard tensor (rather than vector) notation. The nonlinear term is now written as

$$u_j \frac{\partial u_i}{\partial x_j}.$$
* * *
***Reviewer statement 2***: *Furthermore, $\nabla$ is also applied to tensors ($\boldsymbol{\tau}$ and $\boldsymbol{\tau}^{\mathrm{SGS,dev}}$) as $\nabla \cdot \boldsymbol{\tau}$ and $\nabla \cdot \boldsymbol{\tau}^{\mathrm{SGS,dev}}$, while the operation $\nabla\cdot$ for tensors is generally not defined. Besides this, terms $\nabla \cdot \boldsymbol{\tau}$ and $\nabla \cdot \boldsymbol{\tau}^{\mathrm{SGS,dev}}$ (whatever they mean) must enter (2) with the same sign, as $\boldsymbol{\tau}$ and $\boldsymbol{\tau}^{\mathrm{SGS,dev}}$ are quantities of the same physical nature (kinematic stresses). But in fact, concluding from how these variables are defined, $\boldsymbol{\tau} = -2\nu\mathbf{S}$ and $\boldsymbol{\tau}^{\mathrm{SGS,dev}} = -2\nu^{\mathrm{SGS}}\mathbf{S}$, they are rather kinematic momentum fluxes (negative of stresses, cf. Eqs. 5 and 6).*

**Response**: We thank the reviewer for his comment. We have now fixed the sign convention by introducing a minus sign in the viscous term definition. We however decided to keep these terms defined as *stresses*, since this is what they are from physical perspective, irrespective of the sign, and since this is also how they are typically referred to in the literature (see e.g. Pope (2000)). With regards to the divergence operator applied to a second order tensor, we respectfully disagree with the reviewer's comment since

$$\nabla \cdot \boldsymbol{\tau} = \boldsymbol{e}_k \frac{\partial}{\partial x_k}[\boldsymbol{e}_i \boldsymbol{e}_j \tau_{ij}] = \delta_{ki} \boldsymbol{e}_j \frac{\partial \tau_{ij}}{\partial x_k} = \boldsymbol{e}_j \frac{\partial \tau_{ij}}{\partial x_i} \ .$$

Note however that the equations have now been rewritten in index notation with the momentum-flux divergence term taking its standard form (see Eq. 1 in the revised manuscript).
* * *
***Reviewer statement 3***: *Another problem I have is related to the choice of notation for horizontal velocity vector in (4). Notation* **u** *is already reserved for the 3D filtered velocity vector in Eqs. 1 and 2, which implies that* $|\mathbf{u}|$ *can be nothing else than* $\sqrt{u^2 + v^2 + w^2}$, *so it would be necessary to introduce a horizontal 2D velocity vector* $\mathbf{v} = (u, v)$ *with* $|\mathbf{v}| = \sqrt{u^2 + v^2}$ *and make corresponding adjustments in the remaining equations of Sect. 2.1.*

**Response**: The quantity $|\tilde{\mathbf{u}}| \equiv \sqrt{u^2 + v^2}$ has now been defined.
* * *
**1.2 Minor comments**
* * *
***Reviewer statement 1***: *Table 1. Total number of grid/cell points (e.g., $160^3$) is not a measure of grid resolution.*

**Response**: In Table 1, "grid resolution" is replaced by $N_x \times N_y \times N_z$.
* * *
***Reviewer statement 2***: *Figure 1. Primes (') are typically reserved for denoting fluctuations, not their RMS values. May be a source of confusion.*

**Response**: The RMS values of streamwise, cross-stream and vertical velocity fluctuations are now denoted by $u'_{\mathrm{RMS}}$, $v'_{\mathrm{RMS}}$ and $w'_{\mathrm{RMS}}$, respectively.
* * *
***Reviewer statement 3***: *Table 2. Are you sure that you need four decimal places to characterize the relative error?*

**Response**: The relative error is now characterized up to the second decimal place.
* * *
***Reviewer statement 4***: *Table 3. Same problem. Here you even use five decimal places...*

**Response**: Please see the response to statement 3.

**Reviewer statement 5**: *Table 4. See two previous points.*

**Response**: Please see the response to statement 3.
* * *
**Reviewer statement 6**: *Line 294. You need to be more specific about the way double-primed quantities have been evaluated and comment on meaning of their signs.*

**Response**: We now refer to these quantities as "conditionally-averaged velocity fluctuations" and no additional notation is introduced. The sign approach is standard, i.e. a positive velocity fluctuation with respect to the conditionally-averaged mean will have a positive sign, and viceversa for a negative fluctuation with respect to the conditionally-averaged mean.
* * *
**Reviewer statement 7**: *Line 321. Apparently, it should be $C_{\mathrm{S}} = 0.1678$.*

**Response**: The value for the Smagorinsky constant at line 321 is now $C_{\mathrm{S}} = 0.1678$.